# Knowledge Unlearning for Mitigating Privacy Risks in Language Models

## Abstract

Pretrained Language Models (LMs) memorize a vast amount of knowledge during initial pretraining, including information that may violate the privacy of personal lives and identities. Previous work addressing privacy issues for language models has mostly focused on data preprocessing and differential privacy methods, both requiring re-training the underlying LM. We propose *knowledge unlearning* as an alternative method to reduce privacy risks for LMs *post hoc*. We show that simply applying the unlikelihood training objective to target token sequences is effective at forgetting them with little to no degradation of general language modeling performances for larger LMs; it sometimes even substantially improves the underlying LM with just a few iterations. We also find that *sequential* unlearning is better than trying to unlearn all the data at once and that unlearning is highly dependent on which kind of data (domain) is forgotten. By showing comparisons with a previous data preprocessing method and decoding method known to mitigate privacy risks for LMs, we show that unlearning can give a strong empirical privacy guarantee in scenarios where the data vulnerable to extraction attacks are known a priori while being orders of magnitude more computationally efficient and robust. We release the code and dataset needed to replicate our results at http://www.omitted.link/.

## 1 Introduction

Recent work has shown that an adversary can extract training data from Pretrained Language Models (LMs) including Personally Identifiable Information (PII) such as names, phone numbers, and email addresses, and other information such as licensed code, private clinical notes, and 128-bit UUIDs (Carlini et al., 2021; Lee et al., 2022; Huang et al., 2022; Lehman et al., 2021). In 2021, an AI chatbot *Iruda* became the first AI system to be sued for violating the Personal Information Protection Act after generating the exact home addresses and bank account numbers of actual individuals unintentionally (Park, 2021). Heikkilä (2022) has also shown that GPT-3 (Brown et al., 2020), one of the most well known LM currently in commercial use, offered detailed private information about the Editor-in-Chief of MIT Technology Review including his family members, work address, and phone number. Considering findings that show extracting training data gets easier as LMs scale to larger sizes (Carlini et al., 2022a) and that it is common practice for practitioners to release billion parameter pretrained LMs for public use (Gao et al., 2020; Black et al., 2021; Zhang et al., 2022), it has become important to provide privacy guarantees for large LMs.

Practitioners are required to *delete* personal information from the LMs by individuals' request because each individual has the "Right To Be Forgotten (RTBF)" (Mantelero, 2013; Graves et al., 2021) and can limit the direct and indirect commercial use of their personal information (Villaronga et al., 2018). Previous methods addressing privacy risks for language models attempt to remove all private information from the training data (data preprocessing) (Aura et al., 2006; Dernoncourt et al., 2017; Lison et al., 2021; Kandpal et al., 2022) or attempt to design algorithms that ensure differential privacy (DP) (Dwork, 2008; Dwork et al., 2006; Abadi et al., 2016; Anil et al., 2021; Li et al., 2022; Yu et al., 2022). Both approaches require *retraining* the underlying LM every time individuals want to practice their RTBF, which makess them inadequate for large LMs that are extremely costly to retrain. Furthermore, as pointed out by Brown et al. (2022), data preprocessing methods assume private information to be easily identifiable, specified, and removed and DP algorithms can only

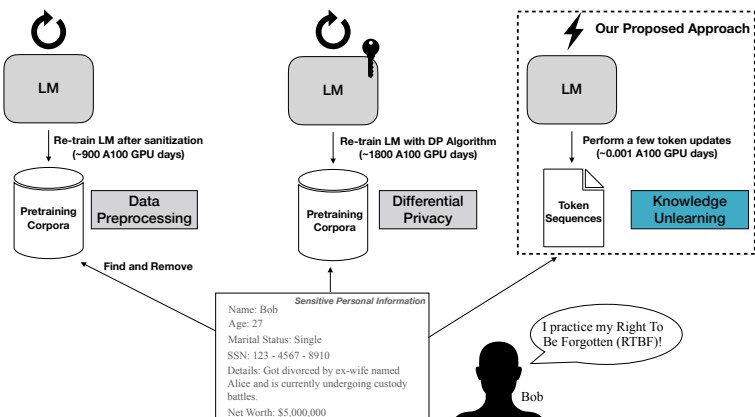

Figure 1: Comparison of previous approaches and *knowledge unlearning* when an individual practices his/her Right-To-Be-Forgotten (RTBF).

guarantee protection for information that has clear privacy borders, which make them inadequate in the real-world scenarios where the standard of privacy might differ by each individuals.

To this end, we propose *knowledge unlearning* (Figure 1) as an efficient solution that can be applied with just a few parameter updates instead of pretraining the underlying LM again. We perform experiments on GPT-Neo LMs (125M, 1.3B, 2.7B) (Black et al., 2021) and show that simply changing the gradient descent to the opposite direction during language modeling (which can also be seen as *maximizing* instead of *minimizing* the loss function) is effective at protecting target sequences from extraction attacks with little to no performance degradation on the initial LM capabilities measured via 9 common NLP classification benchmarks (Hellaswag (Zellers et al., 2019), Lambada (Paperno et al., 2016), Winogrande (Sakaguchi et al., 2021), COPA (Gordon et al., 2012), ARC-Easy (Clark et al., 2018), ARC-Challenge (Clark et al., 2018), Piqa (Bisk et al., 2020), MathQA (Amini et al., 2019), and PubmedQA (Jin et al., 2019)) and 4 dialogue tasks (Wizard of Wikipedia (Dinan et al., 2019), Empathetic Dialogues (Rashkin et al., 2019), Blended Skill Talk (Smith et al., 2020), and Wizard of Internet (Komeili et al., 2022)). For some cases, *knowledge unlearning* unexpectedly shows significant improvements in LM performance for some of the benchmarks.

We compare our approach with data deduplication method (Kandpal et al., 2022) and differential privacy decoding method (Majmudar et al., 2022) which are both known to mitigate privacy risks, and show the effectiveness of knowledge unlearning by providing a strong privacy protection while being much more efficient and robust. We also provide a general guideline that can be used to quantify the *memorization* and *extraction likelihood* of target token sequences and suggest when we can empirically consider them to have been "forgotten". Specifically, we introduce a novel metric that measures the extraction likelihood by varying the prefix length of the target token sequence and quantifying how much of the suffix is actually extracted from the LM.

Surprisingly, for *knowledge unlearning*, we find that it is easier to forget a chunk of instances *sequentially* rather than trying to forget them all at once. We provide further analysis and show that the difficulty of *knowledge unlearning* depends heavily on the target data being forgotten, especially the domain of the target data. We also provide empirical examples of performing extraction attacks and how exactly *knowledge unlearning* provides a privacy protection for the LM.

To summarize, our main contributions are fourfold:

- We compare *knowledge unlearning* with two approaches from literature known to mitigate privacy risks: a data preprocessing approach and a Differential Privacy (DP) Decoding approach. We show that our approach results in little to no performance degradation of general capabilities (sometimes resulting in improvement) while providing a strong privacy protections in situations individuals practice their RTBF whereas the data preprocessing approach provides a weaker privacy protection while being orders of magnitude computationally demanding and the DP Decoding approach results in a severe degradation of modeling performance.

- We perform additional experiments to determine which factors contribute to the difficulty of knowledge unlearning and find that (1) trying to forget many samples at once results in substantial LM performance degradation which can be mitigated by *sequentially* forgetting chunks of data and that (2) the domain of the target data (Code, License, Wikipedia, etc.) plays a critical role in determining how hard they are to forget.

- We provide a novel metric and a general guideline for quantifying the privacy risks for LMs and determine when they should be considered to have "forgotten" a given target sequence.

- *Knowledge unlearning* surprisingly seems to make LMs stronger where the extreme cases bring *+8.0%* (37.6% → 45.6%), *+10.1%* (57.4% → 67.5%), and *+7.9%* (62.2% → 70.1%) improvements on Lambada for GPT-NEO 125M, 1.3B, and 2.7B, respectively.

## 2 RELATED WORK

### 2.1 PRIVACY METHODS FOR LANGUAGE MODELS

Prior work that tries to mitigate privacy risks for LMs can be divided mainly into data pre/post-processing methods and differential privacy methods.

**(Data) Pre/Post-Processing**  Data preprocessing aims to sanitize the training data; it aims to get rid of all data that might violate any kind of privacy from the training data prior to training. These methods mostly utilize measures such as parsers and classification models that try to identify and predict patterns that constitute private information. This is effective at identifying well-formatted private information such as social security numbers or special forms of medical notes (Aura et al., 2006; Dernoncourt et al., 2017; Lison et al., 2021; Kandpal et al., 2022). However, as pointed out by Brown et al. (2022), considering that private information is mostly context-dependent and sometimes in a non-specific format, data preprocessing methods cannot fully claim that they provide privacy guarantees, especially guarantees that match each individual's standards. Methods that attempt to utilize *post-processing* methods such as applying censorship to the LM outputs still face the same limitations.

In this work, we compare our proposed method with a data preprocessing approach proposed by Kandpal et al. (2022) which shows that deduplicating the training corpora before pretraining helps pretrain LMs that show stronger robustness against extraction attacks than an LM pretrained under the same circumstances without deduplicating the pretraining corpora. However, we highlight that this approach, which may still be effective at mitigating the overall privacy risks, is not the most suitable approach when considering a realistic scenario of individuals requesting the removal of their information from the implicit parameters of the LMs.

**Differential Privacy**  Differential Privacy (DP) aims to guarantee that the effect of an individual input on the output of a specific function is bounded (Dwork, 2008; Dwork et al., 2006). In the context of deep neural networks, DP, which needs to be applied during the training phase, aims to construct models that can provide *general* guarantees that the individual information within the training data cannot be inferred (Abadi et al., 2016). While DP has shown to be surprisingly effective at fine-tuning LMs (Li et al., 2022; Yu et al., 2022), pretraining LMs with DP still suffers from substantial performance gap, expensive computation, and slow convergence (Anil et al., 2021). Furthermore, as pointed out by Brown et al. (2022), DP can only provide limited guarantees for LMs because DP requires a unified definition for privacy boundaries, which is inherently impossible for natural language data. Most importantly, in a realistic scenario where individuals may practice their Right-To-Be-Forgotten (RTBF) dynamically after model deployment, it is nontrivial to apply existing descent-based DP algorithms such as DP-SGD to only protection against *targeted* extraction attacks.

### 2.2 MACHINE UNLEARNING

Machine unlearning has received attention as an alternative approach to overcome data privacy issues in machine learning (Cao & Yang, 2015; Ginart et al., 2019; Bourtoule et al., 2021; Graves et al., 2021). Several studies attempt to explore machine unlearning for deep neural networks (Golatkar et al., 2020; Mehta et al., 2022). However, they mostly focus on proposing algorithms for image

classification models where they aim to forget a whole class; that is, achieve random performance for specific image classes such as "cats" or "ships". We are the first, to the best of our knowledge, to explore unlearning a specific sequence of tokens for LMs which is a quite different set-up from traditional image classification models ($\sim$tens of image classes vs. a sequence of tokens that can each be classified into $V \in \mathbb{R}^{\sim 50,000}$). In this work, we coin this approach as *knowledge unlearning* since we are more focused on forgetting specific *knowledge* represented by sequences of tokens.

Zhou et al. (2022) focus on how *forgetting* can be leveraged to improve the performance of the underlying model. They propose "forget-and-relearn" that unifies existing iterative training algorithms by selectively removing undesirable information and re-learning good features, helping boost performance for the task of image classification and multi-agent emergence communication. The underlying assumption is that it is often easier to define and stop unwanted behavior than to teach good behavior. We also show this phenomenon in Section 4 where we unintentionally find unlearning just a few sequences of tokens sometimes boosts general LM capabilities.

### 2.3 MEMORIZATION IN LANGUAGE MODELS

Previous work that explores to which extent LMs have memorized their training data approach the phenomenon with two different viewpoints. Some work view memorization of LMs simply as a threat to individual privacy (Carlini et al., 2021; 2022a; Jagielski et al., 2022) and utilize metrics that quantify how much the LMs are susceptible to adversarial attacks. These metrics are mostly dependent on the specific types of attacks such as the membership inference attack (Shokri et al., 2017) and measure the privacy risks of LMs by quantifying the success rate of these attacks. In our work, we instead focus on more *targeted* extraction attacks.

Another line of work simply quantifies how much *knowledge* is accumulated and forgotten during pretraining by extracting relational knowledge about the world (Petroni et al., 2019; Lazaridou et al., 2021; Jang et al., 2022b;a). This line of work does not view memorization as a negative trait, but as a positive one that can be leveraged to extract world knowledge from its implicit parameters and perform knowledge-intensive tasks such as question answering or training knowledgeable conversation agents.

Our work is highly related to Jagielski et al. (2022)'s work where they also assert that forgetting can be a relaxed version of differential privacy. However, there are two main differences between our work and theirs. First, they only analyze forgetting as a *passive* form of mitigating privacy, asserting that data seen early in large-scale training obtain privacy benefits, whereas we suggest a more *active* form of forgetting. Second, they only show analysis results with image classification and audio generation models while we specifically focus on large LMs.

## 3 KNOWLEDGE UNLEARNING FOR LANGUAGE MODELS

### 3.1 METHODOLOGY

We propose simply *negating* the original training objective of minimizing the negative log-likelihood of the token sequences as our main method of knowledge unlearning in LMs. Specifically, given a sequence of tokens $\boldsymbol{x} = (x_1, ..., x_T)$, our unlearning training objective is simply *maximizing* the following loss function:

$$\mathcal{L}_{UL}(f_\theta, \boldsymbol{x}) = -\sum_{t=1}^{T} \log(p_\theta(x_t | x_{<t}))\tag{1}$$

where $x_{<t}$ denotes the token sequence $x = (x_1, ..., x_{t-1})$ and $p_\theta(x_t | x_{<t})$ denotes the conditional probability of predicting the next token to be $x_t$ when given $x_{<t}$ to an LM $f$ with parameters $\theta$.

Prior work refer to this training objective as *unlikelihood* training and combines it together with the original loss of minimizing the negative log-likelihood for the final objective of enhancing language modeling quality (Welleck et al., 2020) and few-shot learning for downstream NLP tasks (Tam et al., 2021). In contrast, we simply optimize the unlikelihood training objective since we are only concerned with forgetting. While this method seems simple, it is highly effective at forgetting specific token sequences without affecting the overall LM capabilities as shown in Section 4.

### 3.2 Quantifying Privacy Risks of Language Models

In this subsection, we introduce two metrics we use to quantify the privacy risks given a specific token sequence and how we empirically define the token sequence to be forgotten. In this work, we do not utilize metrics such as membership inference attack recall (Shokri et al., 2017) since we are not interested in quantifying the *general* privacy risks of LMs, but instead the privacy risks on the specific target token sequences.

**Extraction Likelihood (EL)** We first introduce a new metric, EL. Given a sequence of tokens $\boldsymbol{x} = (x_1, ..., x_T)$ and an LM $f$ with pre-trained parameters $\theta$, we define EL to be as follows:

$$\text{EL}_n(\boldsymbol{x}) = \frac{\sum_{t=1}^{T-n} \text{OVERLAP}_n(f_\theta(x_{<t}), x_{\geq t})}{T - n} \tag{2}$$

$$\text{OVERLAP}_n(\boldsymbol{a}, \boldsymbol{b}) = \frac{\sum_{c \in n\text{-}grams(\boldsymbol{a})} \mathbb{1}\{c \in n\text{-}grams(\boldsymbol{b})\}}{|n\text{-}grams(\boldsymbol{a})|} \tag{3}$$

where $n\text{-}grams()$ denotes the list of $n$-grams in the given token sequence and $f_\theta(x_{<t})$ denotes the output token sequences from the LM $f_\theta$ when given $x_{<t}$ as input that can have max lengths $|x_{\geq t}|$ but may be shorter when the EOS (end-of-sequence) token is generated beforehand.

The process of varying the prefix length $|x_{<t}|$ can be seen as varying the *strength* of adversarial attacks. This is based on the assumption that the more prior information is provided about the target token sequence, the easier the LM will be able to extract it. Overall, EL can be seen as estimating the general *extraction likelihood* since we are measuring the average success rate of varying extraction attacks quantified via getting the n-gram overlap of generated and target token sequences. While previous metrics quantifying the privacy risks of LMs are dependent on specific adversarial attacks, this characteristic of EL allows it to quantify the general likelihood of extraction without any dependency on specific extraction attacks.

We regard $n$ to be a hyper-parameter that can be varied depending on the stringency of privacy standards. The higher $n$ is set, the stricter we set the standard for a successful extraction attack.

**Memorization Accuracy (MA)** We define Memorization Accuracy (MA) as follows:

$$\text{MA}(\boldsymbol{x}) = \frac{\sum_{t=1}^{T-1} \mathbb{1}\{\arg\max(p_\theta(\cdot|x_{<t})) = x_t\}}{T - 1} \tag{4}$$

MA quantifies how much $f_\theta$ has memorized the given token sequences and was proposed by Tirumala et al. (2022) to analyze the training dynamics of large LMs.

**Empirical Definition of Forgetting** By utilizing both $\text{EL}_n$ and MA, we empirically define a specific token sequence $\boldsymbol{x}$ to be forgotten and is no longer susceptible to extraction attacks when the following conditions are met:

$$\text{EL}_n(\boldsymbol{x}) \leq \frac{1}{|D'|} \sum_{\boldsymbol{x}' \in D'} \text{EL}_n(\boldsymbol{x}') \text{ and } \text{MA}(\boldsymbol{x}) \leq \frac{1}{|D'|} \sum_{\boldsymbol{x}' \in D'} \text{MA}(\boldsymbol{x}') \tag{5}$$

where $D'$ represents a validation corpora not seen during training. In other words, we define $\boldsymbol{x}$ to be forgotten when the $\text{EL}_n(\boldsymbol{x})$ and $\text{MA}(\boldsymbol{x})$ reach a value that is lower than the average $\text{EL}_n$ and MA on token sequences that were not seen during training.

## 4 Experiments

### 4.1 Models, Datasets, and Configurations

**Baselines** For the experiments, we use the GPT-Neo (125M, 1.3B, 2.7B) LMs (Black et al., 2021) initially pretrained on all of the Pile corpora (825GB) (Gao et al., 2020), and the OPT (125M, 1.3B, 2.7B) LMs (Zhang et al., 2022), pretrained on a subset of the *deduplicated* version of the Pile as well

as other corpora from different domains. For the experiments, we perform unlearning the GPT-NEO LMs and quantify the privacy risks of the target data compared to the OPT LMs to measure how effective our proposed approach is in contrast to deduplicating the training corpora before pretraining the underlying LM Kandpal et al. (2022). We do not use the exact LMs from Kandpal et al. (2022) because the LMs were not open-sourced, and thus use the OPT LMs instead. We also consider the Differential Privacy (DP) Decoding (Majmudar et al., 2022) as one of the baselines; This approach proposes a decoding strategy that performs linear interpolation of the original logits with the uniform distribution and performs nucleus sampling, which they theoretically show provides DP guarantees. $\lambda$ is set as the linear interpolation weight where $\lambda = 0$ performs nucleus sampling from the uniform distribution and $\lambda = 1$ performs regular nucleus sampling, using the logits as weights during random sampling.

**Target Data** For the actual target data used to quantify the privacy risks of the LMs, we sample instances from the Training Data Extraction Challenge [1] where 15,000 examples (each are 200 token sequences long) from 16 different domains of the Pile corpora that are identified to be somewhat easy-to-extract are provided. For our experiments, we randomly sample $s$ samples from the 15,000 examples and make the underlying LM forget the $s$ samples at once. As a default, we show the average results of 5 random samplings of $s$ samples for all of our experimental settings. We only provide the average of the 5 samplings and do not separately report the standard deviation. Instead, we provide the results of each individual run in Appendix A.

**Evaluation Datasets** Provding stronger privacy protections for LMs may become meaningless if it requires sacrificing their original capabilities. Thus, while quantifying the privacy risks of LMs, we also quantify the original LM capabilities by evaluating the LMs on 9 different classification tasks quantifying the general capabilities: Hellaswag (Zellers et al., 2019) and Lambada (Paperno et al., 2016) benchmarks to measure linguistic reasoning abilities, Winogrande (Sakaguchi et al., 2021) and COPA (Gordon et al., 2012) to measure commonsense reasoning abilities, and ARC-Easy (Clark et al., 2018), ARC-Challenge (Clark et al., 2018), Piqa (Bisk et al., 2020), MathQA (Amini et al., 2019), PubmedQA (Jin et al., 2019) benchmarks to measure the scientific reasoning abilities. We also evaluate on 4 dialogue tasks (Wizard of Wikipedia (Dinan et al., 2019), Empathetic Dialogues (Rashkin et al., 2019), Blended Skill Talk (Smith et al., 2020), and Wizard of Internet (Komeili et al., 2022)) to evaluate the generation capabilities of the LMs. We use the test set for Lambada and the validation set for the rest of the datasets. We also show the results of measuring the perplexity on the validation corpora of Pile and Wikitext in Appendix B. We do not include measuring perplexity as one of the main evaluations because perplexity might not be the most suitable metric for quantifying general LM performance, especially in the case of unlearning (further explanation given in Appendix B. We evaluate DP Decoding only on the 4 dialogue tasks because the decoding strategy cannot be applied for performing the classification tasks which is evaluated by utilizing a *verbalizer*.

**Configurations** For the learning rate, we set it to 5e-5. We show the effect of varying learning rates in Appendix D. We use a constant learning rate scheduling throughout the run. We fix the global batch size to be the same as $s$ (how many samples are forgotten at once) because having global batch sizes smaller than $s$ proved to degrade general LM capabilities [2]. For $EL_n$, we set $n$=10 which means EL measures the extraction likelihood of extracting $n$ consecutive tokens of varying extraction attack [3]. For calculating $EL_{10}$ and MA, we use a naïve greedy decoding strategy. We set both the dropout and weight decay rates to 0. Lastly, while we provide a guideline of empirically deciding a single token sequence to be forgotten in Section 3.2, for considering a *chunk* of $s$ token sequences to be forgotten, we use the average $EL_{10}$ and MA as an approximation of the individual $EL_{10}$ and MA.

## 4.2 MAIN EXPERIMENTS

**Forgetting Threshold** First, we show how we get the Forgetting Threshold for $EL_{10}$ and MA, the values where we consider the token sequence to be forgotten and unsusceptible from extraction

---

[1] https://github.com/google-research/lm-extraction-benchmark

[2] In Section 4.3, We show that $s$ plays a critical role in determining how much the unlearning will degrade in general capabilities of the LM since $s = 128$ shows to result in much degradation. Method to mitigate this is proposed in Section 4.3 as well.

[3] We set the $n$ value to 10 since we empirically consider an extraction to be successful when 10 consecutive token sequences are successfuly generated by the LM. We show varying the $n$ with values from [5,10,20,40] in Appendix H.

Table 1: Forgetting Threshold for GPT-NEO LMs

| Model (Size) | $EL_{10}$(%) Threshold | MA(%) Threshold |
|---|---|---|
| GPT-NEO (125M) | 4.99 | 29.94 |
| GPT-NEO (1.3B) | 5.68 | 33.27 |
| GPT-NEO (2.7B) | 5.53 | 34.02 |

Table 2: Main Results showing the average of 5 random sampling of $s = 32$ (forgetting 32 samples at once). OPT represents the LM with deduplication applied. NEO denotes the initial GPT-NEO LM, NEO + $DPD^+$ represents applying the DP Decoding strategy by varing the $\lambda$ to match the forgetting criteria, NEO + UL represents performing unlearning on the initial NEO until it provides a stronger security for the target sequences than OPT, NEO + $UL^+$ represents performing unlearning on GPT-NEO until target sequences match the forgetting criteria, **LM Avg.** denotes the average accuracy of the 9 classification datasets, and **Dialogue Avg.** denotes the average F1 score of the 4 dialogue datasets. Best comparable performances are **bolded** and second best underlined.

| Model | # Params | $EL_{10}$ (%)↓ | MA (%)↓ | LM Avg. (ACC)↑ | Dialogue Avg. (F1)↑ | Epoch |
|---|---|---|---|---|---|---|
| OPT | 125M | 8.6 | 52.9 | 42.4 | **10.2** | - |
| NEO | 125M | 30.9 | 77.4 | **43.4** | 9.4 | - |
| NEO + $DPD^+$ | 125M | **0.0** | **27.4** | N/A | 7.3 | - |
| NEO + UL | 125M | 3.7 | 50.1 | 42.6 | 8.0 | 11.0 |
| NEO + $UL^+$ | 125M | 1.0 | **27.4** | 39.9 | 2.6 | 17.2 |
| OPT | 1.3B | 23.3 | 67.1 | **50.6** | **12.4** | - |
| NEO | 1.3B | 67.6 | 92.2 | 49.8 | 11.5 | - |
| NEO + $DPD^+$ | 1.3B | **0.0** | **21.4** | N/A | 7.1 | - |
| NEO + UL | 1.3B | 11.0 | 62.2 | 49.7 | 11.6 | 8.0 |
| NEO + $UL^+$ | 1.3B | 1.9 | 30.4 | 49.7 | 8.5 | 13.8 |
| OPT | 2.7B | 25.6 | 69.2 | **52.7** | **12.9** | - |
| NEO | 2.7B | 70.4 | 93.4 | 52.3 | 11.5 | - |
| NEO + $DPD^+$ | 2.7B | **0.0** | **24.2** | N/A | 6.9 | - |
| NEO + UL | 2.7B | 13.0 | 66.0 | 52.3 | 12.5 | 5.4 |
| NEO + $UL^+$ | 2.7B | 1.6 | 31.0 | 51.9 | 11.1 | 10.8 |

attacks, for all model sizes of GPT-NEO LMs in Table 1. For $D'$, we perform weighted sampling (same domain distribution as the Pile training corpora) of 10,000 instances each with token lengths 200 from the Pile validation corpora and measure the average $EL_{10}$ and MA (Equation 5), which are empirically set as the Forgetting Threshold values.

**Main Results**    Table 2 shows the main results of performing unlearning on LMs of varying sizes and the baselines. While we provide the average performances of the 5 random samplings in Table 2, we provide each individual runs in Appendix A for reference.

We highlight five main observations regarding the results. (1) OPT LMs show a much lower $EL_{10}$ and MA than GPT-NEO LMs, confirming that deduplicating the pretraining corpora is indeed helpful for mitigating privacy risks. (2) NEO + $DPD^+$ enables effective protection against extraction attacks demonstrated via the lowest EL and MA score; however, it brings severe degradation of generation capabilities measured via the Average F1 score of the 4 dialogue generation tasks. (3) NEO + $UL^+$ results in severe degradation of both classification and dialogue tasks for the 125M, only severe degradation of dialogue tasks for 1.3B LM while for the 2.7B LMs, it enables retaining most of its previous capabilities. (4) While the LMs scale to larger sizes, it takes fewer epochs for the target sequences to be forgotten. Together with (3), this implies that larger LMs are strong *un*learners. (5) While NEO + $UL^+$ provides a stronger privacy protection than OPT without sacrificing its performance from NEO for the 2.7B LM, it is much more computationally efficient (3,500,000x) than re-training the underlying LM, which is required for all data preprocessing approaches [4].

---

[4] Computational efficiency is measured via FLOPs which is calculated by (6 × Total Training Tokens × Parameter Size) as in Brown et al. (2020). FLOPs for OPT LMs were estimated using information from Zhang et al. (2022). We provide the FLOPs for the methods in Appendix C.

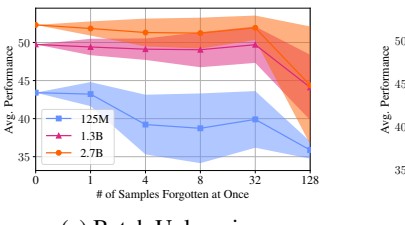
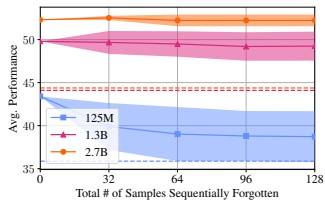

| (a) Batch Unlearning | (b) Sequential Unlearning |

Figure 2: Average LM performance on the 9 classification benchmarks when varying the total number of samples forgotten at once is shown in (a) and the average LM performances when the 128 samples are divided into 4 chunks and are forgotten sequentially is shown in (b). The lines denote the average performances of 5 random samplings and the standard deviation is shown as the shaded regions. The dotted lines in (b) denotes the $s = 128$ performance in (a) for comparison purposes.

Overall, results show unlearning to be an effective approach to providing a strong privacy protection while retaining and sometimes even improving general LM capabilities.

**Sequential Unlearning is more Stable than Batch Unlearning.** We show the effect of varying $s$ (the # of data instances to be forgotten at once) in Figure 2a across model scales. We denote this approach as *batch* unlearning. As shown by the $s = 128$ results, it is harder to forget more samples at once, resulting in substantial degradation of average LM performance regardless of how large the LM is. Since $s \leq 32$ does not show much degradation, we explore if *sequentially* unlearning can be a solution. In Figure 2b, we show the result of dividing the 128 samples into 4 chunks of 32 and performing sequential unlearning; we unlearn each chunk at a time until the chunk reaches the forgetting threshold. Surprisingly, as shown by the performance gap at $s = 128$ between the dotted lines (the $s = 128$ performance of Figure 2a) and straight lines, the end result is vastly different even though exactly the same instances were forgotten. Sequential unlearning shows almost no degradation of average LM performance. In Appendix G, we show that chunks once forgotten stay forgotten and that later chunks are forgotten much faster compared to the initial chunk. This result hints at the *generalization* of unlearning, which we do not further explore in the scope of this work. The result also suggests that knowledge unlearning can be *continually* applied to LMs when needed.

### 4.3 ANALYSIS OF KNOWLEDGE UNLEARNING

**Providing Better Intuition of What Exactly Happens During Knowledge Unlearning.** To show exactly what happens to the LM during knowledge unlearning, we show how the performance of each of the LM benchmarks changes as we perform 10 runs of unlearning to the GPT-NEO (1.3B) model (each run with $s = 1$) in Figure 3. As shown in the figure, the LM performance for each benchmark varies tremendously on which sample is chosen to be forgotten. Furthermore, the ending time of each run is different, indicating that some samples are forgotten faster than others. We also show empirical examples of performing actual extraction attacks with prefix length of 100 in Appendix F.

**Towards Understanding Why Some Instances are Harder to Forget** To measure why some instances are harder to forget, we perform 5 random samplings of $s = 8$ from 8 different domains from the Training Data Extraction Challenge [5] and perform unlearning on the GPT-NEO 1.3B LM. We also show the results of each individual run in Appendix A. As shown in Table 3, despite undergoing the same number of token updates (10 epochs of unlearning), different domains result in vastly different outcomes; ENRON EMAILS results in the average LM performance degradation of only -0.4% while USPTO BACKGROUNDS results in -4.5% degradation. Furthermore, the final $EL_{10}$ varies depending on the domain, suggesting that some domains (e.g., FREELAW) are harder to forget than others. Lastly, domains that are more *structured*, which means the data consists of some kind of patterns such as a list of emails (ENRON EMAILS) or code (GITHUB (CODE)), seem to result in less degradation of LM performance in contrast to domains that are more *unstructured*, which means the data consist of mostly raw English text such as a review for journal submission (PUBMED

---
[5]https://github.com/google-research/lm-extraction-benchmark

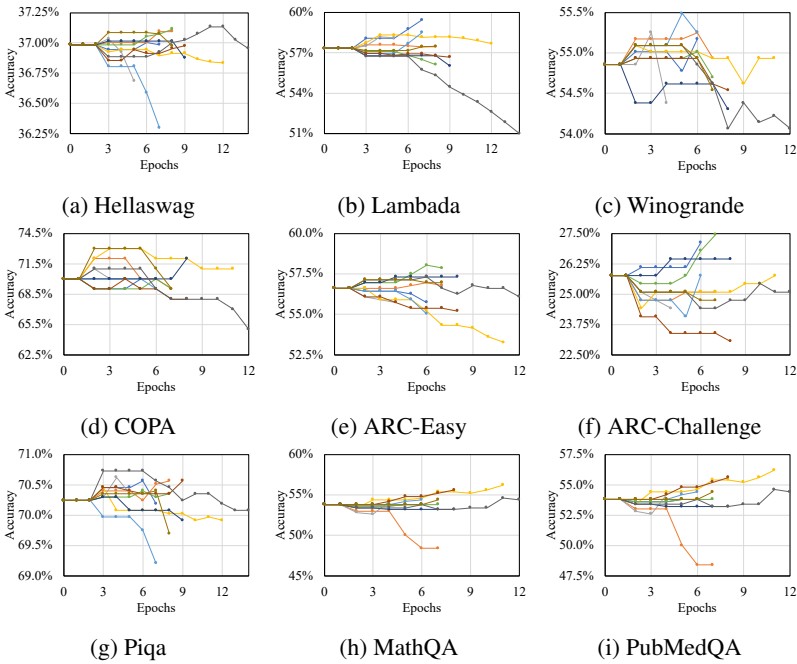

Figure 3: Performance on the 9 classification benchmarks as we perform 10 different unlearning runs on GPT-NEO 1.3B where $s = 1$.

Table 3: Unlearning GPT-NEO 1.3B on token sequences sampled from 8 different domains. We fix the epoch to 10, set $s = 8$ and show the result of the average of 5 random samplings. *Italicized* () denotes the $\Delta$ from INITIAL.

| Domains | Initial $EL_{10}$ | Final $EL_{10}$ | Hella. (ACC) | Lamba. (ACC) | Wino. (ACC) | COPA (ACC) | ARC-E (ACC) | ARC-C (ACC) | Piqa (ACC) | MathQ (ACC) | PubQ (ACC) | Avg. (ACC) |
|---|---|---|---|---|---|---|---|---|---|---|---|---|
| INITIAL | - | - | 37.0 | **57.4** | **54.9** | 70.0 | **56.6** | 25.8 | **70.4** | **21.9** | 53.8 | **49.8** (*0.0*) |
| FREELAW | 60.4 | 12.1 | 37.2 | 52.2 | 53.9 | 68.4 | 55.5 | 26.2 | 70.1 | 21.7 | 53.5 | 48.7 (*-1.1*) |
| GIT. (CODE) | 63.9 | 0.6 | **37.3** | 53.4 | 54.4 | 69.2 | 56.3 | 26.0 | 69.9 | 21.5 | 49.8 | 48.7 (*-1.1*) |
| GIT. (LICENSE) | 75.8 | 0.0 | 37.1 | 52.0 | 54.2 | 69.0 | 56.4 | 26.4 | 70.1 | 21.8 | 51.8 | 48.8 (*-1.0*) |
| ENRON EMAILS | 77.3 | 0.0 | 36.9 | 57.2 | 54.8 | 68.4 | 55.8 | 26.3 | 69.8 | 21.8 | 53.1 | 49.4 (*-0.4*) |
| BOOKS3 | 70.2 | 0.0 | 36.4 | 49.5 | 54.2 | **70.8** | 55.6 | 25.5 | 69.9 | 21.7 | 47.4 | 47.9 (*-1.9*) |
| PILE CC | 67.8 | 0.0 | 35.7 | 45.9 | 53.8 | 70.4 | 54.2 | **26.9** | 69.7 | 21.8 | 52.0 | 47.8 (*-2.0*) |
| USPTO BACK. | 59.4 | 0.0 | 33.7 | 44.7 | 53.5 | 67.0 | 45.9 | 24.0 | 67.0 | 21.5 | 50.3 | 45.3 (*-4.5*) |
| PUBMED CENT. | 71.8 | 0.0 | 36.5 | 44.5 | 54.1 | 69.6 | 55.6 | 24.8 | 70.0 | **21.9** | 46.4 | 47.0 (*-2.8*) |

CENTRAL). We provide examples from each domain in Appendix E. However, further analysis of understanding exactly which components make unlearning work should be made in future work.

## 5 CLOSING

In this paper, we propose *knowledge unlearning* as a method for mitigating privacy risks in LMs that provides a strong privacy protection with little to no degradation of general LM capabilities measured by evaluating on 9 common LM classification benchmarks and 4 dialogue benchmarks for the larger sized LMs. As large LMs expand their use cases, potentially affecting the daily lives of people, the research community should make sure that the privacy of individuals is not violated intentionally or unintentionally by the knowledge stored in the implicit parameters of these models. Since it is inherently impossible to prevent and predict all future privacy concerns prior to pretraining the LM, we suggest the community consider knowledge unlearning for ensuring privacy upon individuals' requests post hoc pretraining. [6]

---

[6] We provide some limitations of our work in Appendix I.

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

## A  Full Results

We provide all of the results for the 5 random samplings for our main experimental setting in Table 4 and the full results for the domain analysis setting in Table 5. We also provide the evaluation of the 4 dialogue tasks for $s = 32$ for all model sizes in Table 6

## B  Measuring Pile and Wikitext Perplexity

Table 7 shows the results of measuring perplexity on 500 samples from the validation set of Pile and Wikitext corpora on the LMs from the main experimental setting (Table 2). Results show that LMs that underwent knowledge unlearning show higher perplexity while the main experimental table (Table 2) does not show degradation of performance on 9 different LM benchmarks. We believe the discrepancy to be due to the inherent attributes of performing unlearning: since we are doing gradient *ascent*, we are likely *softening* the probability to generate each token from the vocabulary, giving it a more uniform distribution that will inevitably result in a higher perplexity. However, since it does not show much degradations in the LM benchmarks, it also means that the $argmax$ of the most likely token to be generated has not changed much. However, further exploration of what exactly *knowledge unlearning* does to the representations of the LM should be done in future work.

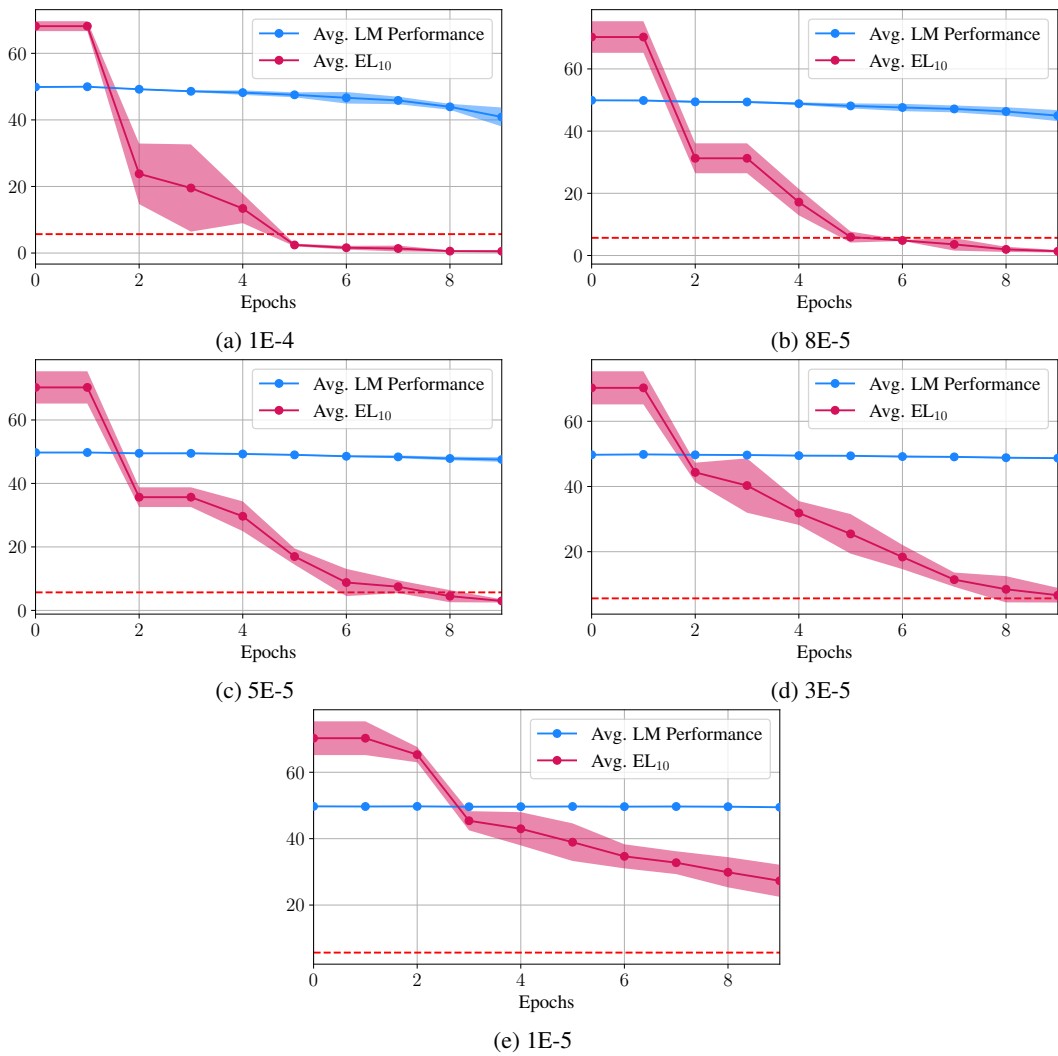

Figure 4: Varying the learning rate for unlearning the GPT-NEO 1.3B with $s = 32$. We report the average of 3 random samplings and display the standard deviations as the shaded regions. Red dotted lines denote the memorization accuracy forgetting threshold of the 1.3B model reported in Table 1.

## C   COMPUTATION COMPARISON BETWEEN DEDUPLICATION AND KNOWLEDGE UNLEARNING

We show the FLOPs of pretraining OPT denoted as DEDUPLICATION and the average FLOPs of performing knowledge unlearning until $s = 32$ token sequences reach the Forgetting Threshold denoted as UNLEARNING in Table 8. We calculate FLOPs by (6 × Total Training Tokens × Parameter Size) following Brown et al. (2020).

## D   VARYING THE LEARNING RATE

In Figure 4, we show the results of varying the learning rate for knowledge unlearning where we fix the total epoch to 10 and perform 3 random runs with $s = 32$ on the GPT-NEO 1.3B. Overall, we observe that higher learning rates lead to faster forgetting, but with substantial LM performance degradation. While lower learning rates retain the LM performance, they fail to meet the Forgetting Threshold within 10 epochs. Thus, we set the learning rate to 5e-5 for our experiments to get the best trade-off.

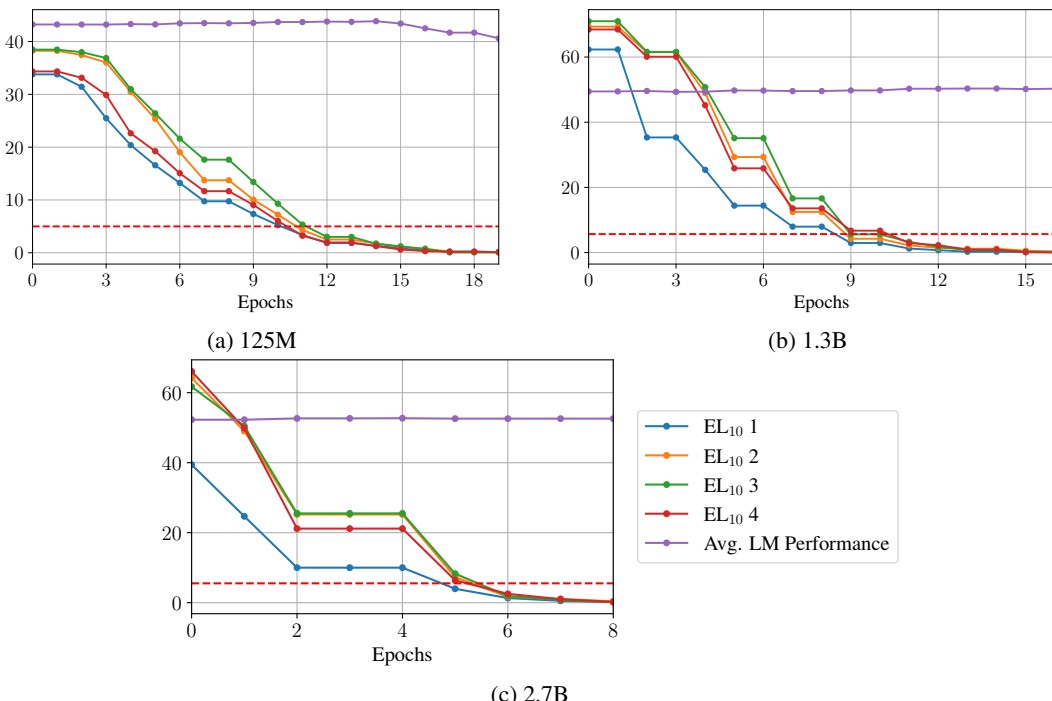

Figure 5: Additional results of sequential unlearning for GPT-NEO 125M, 1.3B, and 2.7B. Red dotted lines denote the memorization accuracy forgetting threshold reported of each model in Table 1.

## E  TEXT EXAMPLE FROM EACH DOMAIN

We show an example token sequence from each of the 8 domains used for the analysis section in Table 9.

## F  MORE EXAMPLES OF PERFORMING EXTRACTION ATTACKS

In addition to the extraction attack example shown in the analysis section, we provide 3 additional examples to provide readers with more empirical examples of how knowledge unlearning ensures protection against extraction attacks in Table 10.

## G  ADDITIONAL RESULTS OF SEQUENTIAL KNOWLEDGE UNLEARNING

We show how the $EL_{10}$ of each individual chunks and the average LM performance change as we perform sequential unlearning in Figure 5. Results show that the chunks that are forgotten stay forgotten and that later chunks are forgotten much faster (one or two epochs) compared to the initial chunk. We hypothesize that this might be because of the similarity of the token sequences from the 15,000 examples from the Training Extraction Challenge Benchmark. Also, this result hints at the *generalization* of unlearning, which we do not further explore because of the scope of this work.

## H  THE EFFECT OF VARYING N FOR EXTRACTION LIKELIHOOD (EL) METRIC

First we show the Extraction Liklihood (EL) Forgetting Threshold values for n=[5,10,20,40] by measuring the value on the 10,000 validation instances unseen during training in Table 11. Next, we show the average LM performance (on the 9 classification benchmarks) where we perform unlearning on the LM on 32 samples until the target token sequences are forgotten (the EL MA value

are both lower than the threshold values) in Table 12. Performance shows the average of 5 random samplings.

## I  LIMITATIONS

While we provide privacy guarantee through unlearning, our Forgetting Threshold is dependent on which data samples are chosen as $D'$. Furthermore, varying the prefix length can be seen as a naïve way of varying the strength of the extraction attacks. In a real-world scenario, extraction attacks may be more complicated and may require other prevention methods. Also, we could not directly compare our approach with a Differential Privacy (DP) (Anil et al., 2021) approach because there are no open-sourced LMs pretrained with a DP algorithm. We could not replicate the pretrainig phase because of the heavy computational resources needed to pretrain an LM with DP which is estimated to require thousands of GPU hours. We leave this comparison for future work. Finally, a recent work (Carlini et al., 2022b) has suggested that machine unlearning (for the vision domain) can bring negative effects harming the privacy of other users. Future work should explore this phenomenon in the setting of performing unlearning on large language models as well.

Table 4: All of the individual runs for the Main Results

| Model (s) | # Params | EL$_{10}$ (%)↓ | MA (%)↓ | Hella. (ACC) | Lamba. (ACC) | Wino. (ACC) | COPA (ACC) | ARC-E (ACC) | ARC-C (ACC) | Piqa (ACC) | MathQ (ACC) | PubQ (ACC) | Avg. (ACC) | Epoch |
|---|---|---|---|---|---|---|---|---|---|---|---|---|---|---|
| NEO | 125M | 30.9 | 77.4 | 28.2 | 37.6 | 51.8 | 62.0 | 45.6 | 22.0 | **63.3** | 22.5 | 57.6 | 43.4 | - |
| Δ | - | - | - | +0.2 | +8.0 | +1.9 | +5.0 | +0.0 | +2.2 | +0.0 | +0.3 | +0.0 | +2.0 | - |
| NEO + UL$^+$ (s = 1) | 125M | 3.1 | 28.1 | 28.1 | 41.0 | 52.5 | 62.0 | 43.2 | 21.0 | 63.0 | **22.8** | 57.6 | 43.5 | 14.0 |
|  | 125M | 0.0 | 27.6 | 28.1 | 24.9 | 50.8 | **67.0** | 42.3 | 23.7 | 62.8 | 21.9 | 57.6 | 42.1 | 10.0 |
|  | 125M | 0.0 | 27.1 | 28.1 | 42.1 | 52.5 | 63.0 | 44.1 | 20.3 | 62.6 | 22.5 | 57.6 | 43.7 | 5.0 |
|  | 125M | 0.0 | 25.6 | 28.2 | 44.9 | 52.0 | 62.0 | 41.8 | 21.4 | 62.6 | 22.2 | 57.6 | 43.6 | 11.0 |
|  | 125M | 0.0 | 28.1 | **28.4** | 33.9 | 51.5 | 66.0 | 44.8 | 21.7 | 62.8 | 22.3 | 57.6 | 43.2 | 10.0 |
| NEO + UL$^+$ (s = 4) | 125M | 0.9 | 28.8 | 27.8 | 44.1 | 51.9 | 52.0 | 37.4 | 19.7 | 60.5 | 22.3 | 57.6 | 41.5 | 16.0 |
|  | 125M | 0.0 | 28.6 | 27.4 | 2.5 | 49.4 | 59.0 | 38.6 | 23.1 | 60.5 | 21.2 | 43.8 | 36.2 | 19.0 |
|  | 125M | 3.6 | 28.8 | 27.7 | 33.4 | 51.8 | 55.0 | 37.7 | 21.0 | 61.0 | 22.3 | 57.6 | 40.8 | 20.0 |
|  | 125M | 2.6 | 28.9 | 27.6 | 29.9 | 52.4 | 50.0 | 36.5 | 19.0 | 60.3 | 22.2 | 57.6 | 39.5 | 18.0 |
|  | 125M | 0.0 | 28.4 | 27.6 | 6.7 | 49.7 | 61.0 | 42.5 | 22.7 | 61.0 | 21.4 | 50.6 | 38.1 | 16.0 |
| NEO + UL$^+$ (s = 8) | 125M | 0.0 | 28.5 | 27.6 | 35.0 | 51.8 | 51.0 | 37.6 | 18.0 | 60.1 | 22.4 | 57.6 | 40.1 | 16.0 |
|  | 125M | 2.2 | 28.1 | 27.7 | 5.4 | 49.6 | 62.0 | 40.6 | 21.0 | 61.2 | 21.8 | 52.4 | 38.0 | 19.0 |
|  | 125M | 0.3 | 29.6 | 28.0 | 41.2 | 52.2 | 55.0 | 40.2 | 21.4 | 61.0 | 21.9 | 57.6 | 42.0 | 18.0 |
|  | 125M | 5.0 | 25.3 | 27.4 | 1.3 | 49.6 | 65.0 | 37.6 | **24.4** | 59.2 | 21.2 | 33.8 | 35.5 | 23.0 |
|  | 125M | 0.0 | 28.2 | 27.9 | 5.3 | 50.5 | 61.0 | 41.6 | 22.4 | 60.7 | 21.5 | 51.4 | 38.0 | 18.0 |
| NEO + UL$^+$ (s = 32) | 125M | 0.3 | 28.4 | 27.2 | 42.3 | **53.7** | 56.0 | 38.1 | 21.0 | 59.7 | 22.4 | 57.6 | 42.0 | 20.0 |
|  | 125M | 0.8 | 27.1 | 27.0 | 17.1 | 52.4 | 53.0 | 34.0 | 20.0 | 59.8 | 21.5 | 57.6 | 38.0 | 18.0 |
|  | 125M | 0.2 | 24.1 | 27.3 | **45.6** | 51.9 | 50.0 | 38.6 | 20.7 | 59.6 | 22.6 | 57.6 | 41.5 | 13.0 |
|  | 125M | 3.0 | 28.7 | 27.5 | 2.6 | 49.2 | 59.0 | 37.7 | 21.4 | 58.4 | 20.9 | 46.8 | 35.9 | 20.0 |
|  | 125M | 0.7 | 28.5 | 27.3 | 44.5 | 53.0 | 54.0 | 39.0 | 20.3 | 59.5 | 22.5 | 57.6 | 42.0 | 15.0 |
| NEO + UL$^+$ (s = 128) | 125M | 1.3 | 28.1 | 27.1 | 4.6 | 50.5 | 58.0 | 37.9 | 21.3 | 57.5 | 21.4 | 47.8 | 36.2 | 16.0 |
|  | 125M | 3.1 | 27.5 | 26.9 | 1.8 | 50.5 | 60.0 | 36.4 | 22.3 | 56.6 | 21.2 | 41.8 | 35.3 | 18.0 |
|  | 125M | 3.9 | 26.7 | 27.0 | 3.9 | 50.9 | 59.0 | 35.2 | 21.3 | 56.0 | 21.3 | 49.6 | 36.0 | 17.0 |
|  | 125M | 2.4 | 26.6 | 26.9 | 2.7 | 50.2 | 56.0 | 35.9 | 22.3 | 57.2 | 21.2 | 43.8 | 35.1 | 16.0 |
|  | 125M | 3.8 | 27.3 | 27.0 | 6.4 | 50.9 | 57.0 | 37.3 | 21.3 | 57.2 | 21.2 | 52.0 | 36.7 | 17.0 |
| NEO | 1.3B | 67.6 | 92.2 | 37.0 | 57.4 | 54.8 | 70.0 | 56.6 | 25.8 | 70.4 | 21.9 | 53.8 | 49.8 | - |
| Δ | - | - | - | +0.4 | +10.1 | +2.1 | +2.0 | +1.1 | +3.4 | +0.3 | +0.4 | +3.8 | +2.6 | - |
| NEO + UL$^+$ (s = 1) | 1.3B | 0.0 | 27.6 | 36.8 | 52.1 | 54.7 | **72.0** | 55.9 | 27.8 | 69.7 | 21.5 | 53.0 | 49.3 | 9.0 |
|  | 1.3B | 0.0 | 30.2 | 36.6 | 54.6 | 54.9 | 69.0 | 55.4 | 21.7 | **70.7** | 21.7 | 53.4 | 49.2 | 6.0 |
|  | 1.3B | 0.0 | 29.7 | 36.7 | 58.2 | 55.4 | 70.0 | 56.1 | 25.4 | 69.9 | 22.0 | 53.2 | 49.7 | 4.0 |
|  | 1.3B | 0.0 | 32.2 | 37.1 | 52.4 | 53.7 | 68.0 | 56.1 | 25.4 | 70.1 | 21.8 | 54.2 | 48.6 | 8.0 |
|  | 1.3B | 0.0 | 27.6 | 37.3 | 60.1 | 55.6 | 70.0 | 57.5 | 25.1 | 70.0 | 21.7 | 55.2 | 50.3 | 10.0 |
| NEO + UL$^+$ (s = 4) | 1.3B | 0.0 | 30.3 | 37.3 | 48.3 | 54.4 | 70.0 | 55.0 | **29.2** | 69.9 | 20.6 | 56.0 | 49.0 | 12.0 |
|  | 1.3B | 0.0 | 29.7 | 36.8 | 49.4 | 53.4 | 69.0 | 55.2 | 26.8 | 70.6 | 21.4 | 52.8 | 48.4 | 9.0 |
|  | 1.3B | 1.0 | 29.2 | 36.8 | 51.3 | 54.9 | 70.0 | 55.2 | 26.8 | 70.3 | 21.5 | 54.0 | 49.0 | 10.0 |
|  | 1.3B | 4.8 | 31.4 | 37.2 | 59.2 | 54.8 | 71.0 | 54.9 | 25.8 | 69.5 | 21.9 | 50.2 | 49.4 | 10.0 |
|  | 1.3B | 1.7 | 31.8 | 37.0 | 58.4 | 54.4 | 71.0 | **57.7** | 24.7 | 70.2 | 22.0 | 54.0 | 49.9 | 9.0 |
| NEO + UL$^+$ (s = 8) | 1.3B | 0.3 | 29.7 | 37.1 | 66.5 | 54.5 | 70.0 | 52.0 | 26.8 | 69.4 | 21.7 | 56.8 | 50.5 | 13.0 |
|  | 1.3B | 1.9 | 29.5 | 36.8 | 43.0 | 53.1 | 71.0 | 51.3 | 27.5 | 70.4 | 21.0 | 42.4 | 46.3 | 13.0 |
|  | 1.3B | 0.2 | 26.2 | 37.2 | 47.3 | 54.2 | 72.0 | 55.2 | 25.8 | 70.4 | 21.8 | 54.8 | 48.7 | 12.0 |
|  | 1.3B | 3.1 | 32.0 | **37.4** | 57.6 | 54.3 | 70.0 | 56.1 | 26.8 | 69.8 | 21.5 | 54.8 | 49.8 | 14.0 |
|  | 1.3B | 1.4 | 32.0 | 37.1 | 57.4 | 54.5 | 71.0 | 57.0 | 26.1 | 70.0 | 21.9 | 54.2 | 49.9 | 11.0 |
| NEO + UL$^+$ (s = 32) | 1.3B | 0.7 | 33.0 | 36.5 | 63.2 | 55.9 | 70.0 | 52.4 | 25.1 | 69.7 | 21.8 | 55.4 | 50.0 | 13.0 |
|  | 1.3B | 1.7 | 29.8 | 36.7 | 50.9 | 53.5 | 71.0 | 56.3 | 27.8 | 70.7 | 22.0 | 39.4 | 47.6 | 14.0 |
|  | 1.3B | 0.7 | 28.4 | 37.0 | 64.8 | **56.9** | 69.0 | 54.3 | 26.4 | 69.1 | 21.9 | **50.6** | 50.6 | 13.0 |
|  | 1.3B | 4.2 | 31.2 | 35.8 | **67.5** | 55.3 | 67.0 | 51.5 | 25.4 | 68.1 | 21.3 | 56.6 | 49.8 | 14.0 |
|  | 1.3B | 2.1 | 29.5 | 35.8 | 63.9 | 55.7 | 70.0 | 54.1 | 26.4 | 69.5 | **22.3** | 56.8 | 50.5 | 15.0 |
| NEO + UL$^+$ (s = 128) | 1.3B | 0.4 | 24.5 | 31.1 | 54.2 | 55.2 | 69.0 | 53.2 | 24.7 | 66.1 | 21.9 | 56.4 | 48.0 | 6.0 |
|  | 1.3B | 4.9 | 19.8 | 27.8 | 2.2 | 54.8 | 69.0 | 50.9 | 23.3 | 57.9 | 21.8 | 55.8 | 40.4 | 8.0 |
|  | 1.3B | 4.2 | 30.2 | 30.6 | 41.6 | 55.1 | 69.0 | 54.4 | 26.0 | 63.8 | 22.1 | 55.0 | 46.4 | 6.0 |
|  | 1.3B | 2.9 | 23.6 | 27.6 | 8.8 | 52.9 | 68.0 | 44.5 | 18.9 | 57.7 | 21.6 | 57.4 | 39.7 | 9.0 |
|  | 1.3B | 1.3 | 23.1 | 28.5 | 48.6 | 55.5 | 69.0 | 48.8 | 21.6 | 62.3 | 22.2 | **57.6** | 46.0 | 8.0 |
| NEO | 2.7B | 70.4 | 93.4 | 40.8 | 62.2 | 56.4 | **75.0** | 59.6 | 25.4 | 73.0 | 21.4 | 57.0 | 52.3 | - |
| Δ | - | - | - | +0.8 | +7.9 | +1.0 | +0.0 | +1.5 | +4.3 | +0.3 | +1.1 | +1.0 | +2.0 | - |
| NEO + UL$^+$ (s = 1) | 2.7B | 0.0 | 3.0 | 40.8 | 62.2 | 56.6 | 72.0 | 55.7 | 26.4 | 73.1 | 21.8 | 57.6 | 51.8 | 10.0 |
|  | 2.7B | 0.0 | 23.6 | 40.5 | 56.8 | 54.4 | 74.0 | 59.6 | 26.1 | 72.8 | 21.3 | 56.6 | 51.3 | 8.0 |
|  | 2.7B | 0.0 | 27.6 | 40.6 | 62.5 | 57.0 | 75.0 | 59.1 | 24.7 | 73.0 | 21.5 | 56.6 | 52.2 | 6.0 |
|  | 2.7B | 0.0 | 20.6 | 40.5 | 60.3 | 55.8 | 74.0 | 58.9 | 25.8 | 73.0 | 21.7 | 57.2 | 51.9 | 10.0 |
|  | 2.7B | 0.0 | 29.7 | 40.6 | 62.2 | 56.4 | 72.0 | 58.0 | 27.1 | 72.2 | 21.2 | 57.4 | 51.9 | 9.0 |
| NEO + UL$^+$ (s = 4) | 2.7B | 0.4 | 22.6 | 41.5 | 60.0 | 54.9 | 72.0 | 55.0 | 26.4 | 69.9 | 21.3 | 57.8 | 51.0 | 12.0 |
|  | 2.7B | 0.0 | 30.0 | **41.6** | 46.5 | 53.4 | 71.0 | 55.6 | 25.1 | 72.0 | 21.3 | 57.2 | 49.3 | 9.0 |
|  | 2.7B | 0.7 | 23.7 | 40.4 | 59.7 | 54.9 | 74.0 | 58.7 | 23.7 | 72.5 | 20.8 | 57.4 | 51.3 | 9.0 |
|  | 2.7B | 3.2 | 32.4 | 41.2 | 67.2 | 56.0 | 73.0 | 57.3 | 28.1 | **73.3** | 22.3 | 57.2 | **52.8** | 8.0 |
|  | 2.7B | 0.2 | 31.9 | 40.3 | 61.2 | 55.7 | 74.0 | 60.0 | 27.5 | 72.0 | 21.4 | 57.2 | 52.1 | 10.0 |
| NEO + UL$^+$ (s = 8) | 2.7B | 0.3 | 29.5 | 41.2 | 64.6 | 55.4 | 71.0 | 52.9 | 27.1 | 69.5 | 21.7 | **58.0** | 51.3 | 10.0 |
|  | 2.7B | 2.1 | 26.4 | 40.6 | 48.7 | 52.9 | 67.0 | 55.0 | 25.8 | 72.1 | 21.8 | 57.2 | 49.0 | 11.0 |
|  | 2.7B | 0.5 | 31.2 | 41.1 | 54.1 | 55.0 | 74.0 | 59.3 | 25.1 | 72.5 | 22.1 | 57.4 | 51.2 | 11.0 |
|  | 2.7B | 1.9 | 33.8 | 40.7 | 65.7 | **57.4** | 72.0 | 58.4 | 27.1 | 72.6 | 21.9 | 57.2 | 52.5 | 8.0 |
|  | 2.7B | 0.0 | 20.4 | 40.0 | 60.7 | 55.8 | 73.0 | 60.1 | 28.5 | 72.5 | 21.5 | 57.2 | 52.2 | 11.0 |
| NEO + UL$^+$ (s = 32) | 2.7B | 0.6 | 31.7 | 40.8 | 68.2 | 56.1 | 68.0 | 54.4 | 28.0 | 71.9 | 21.4 | 57.0 | 51.8 | 11.0 |
|  | 2.7B | 1.1 | 32.4 | 40.9 | 56.9 | 55.6 | 69.0 | 58.1 | 26.7 | 71.8 | 22.1 | 56.8 | 50.9 | 10.0 |
|  | 2.7B | 1.2 | 29.0 | 41.5 | 65.8 | 56.8 | 68.0 | 59.3 | 27.0 | 72.0 | 22.3 | 57.8 | 52.3 | 11.0 |
|  | 2.7B | 3.4 | 29.9 | 39.7 | **70.1** | 57.7 | 68.0 | 54.8 | **29.7** | 71.6 | 22.0 | 57.6 | 52.4 | 11.0 |
|  | 2.7B | 1.9 | 31.9 | 41.4 | 61.6 | 56.6 | 73.0 | **61.1** | 26.4 | 72.7 | 21.7 | 57.0 | 52.4 | 11.0 |
| NEO + UL$^+$ (s = 128) | 2.7B | 0.4 | 31.5 | 35.3 | 64.2 | 56.8 | 68.3 | 51.8 | 26.7 | 70.2 | 21.9 | 56.7 | 50.2 | 10.0 |
|  | 2.7B | 3.8 | 16.5 | 26.0 | 0.4 | 51.6 | 57.7 | 29.0 | 16.6 | 54.2 | 20.0 | 57.9 | 34.8 | 10.0 |
|  | 2.7B | 0.6 | 31.4 | 34.9 | 58.9 | 55.2 | 69.2 | 54.8 | 24.7 | 70.0 | **22.5** | 57.7 | 49.8 | 9.0 |
|  | 2.7B | 2.2 | 31.1 | 31.3 | 22.9 | 50.6 | 62.5 | 40.0 | 18.2 | 60.8 | 21.3 | 40.9 | 38.7 | 8.0 |
|  | 2.7B | 4.7 | 29.0 | 33.5 | 56.5 | 55.0 | 66.3 | 51.9 | 23.6 | 68.6 | 22.4 | 57.7 | 48.4 | 9.0 |

Table 5: All of the individual runs for the Domain Analysis Results for GPT-NEO 1.3B LM.

| Domains | Initial $EL_{10}$ | Final $EL_{10}$ | Hella. (ACC) | Lamba. (ACC) | Wino. (ACC) | COPA (ACC) | ARC-E (ACC) | ARC-C (ACC) | Piqa (ACC) | MathQ (ACC) | PubQ (ACC) | Avg. (ACC) |
|---|---|---|---|---|---|---|---|---|---|---|---|---|
| INITIAL | - | - | 37.0 | 57.4 | 54.9 | 70.0 | 56.6 | 25.8 | 70.4 | 21.9 | 53.8 | 49.8 |
| FREELAW | 64.6 | 4.8 | 37.3 | 53.5 | 54.1 | 68.0 | 57.5 | 27.1 | 70.5 | 21.5 | 54.0 | 49.3 |
| | 52.0 | 2.4 | 37.3 | 62.9 | 54.2 | 67.0 | 52.9 | 26.1 | 69.2 | 21.5 | 54.4 | 49.5 |
| | 60.6 | 15.2 | 36.8 | 42.0 | 54.5 | 67.0 | 56.6 | 25.1 | 70.1 | 21.7 | 51.4 | 47.2 |
| | 55.2 | 13.8 | 37.3 | 51.4 | 53.5 | 69.0 | 55.4 | 26.8 | 70.5 | 21.9 | 54.6 | 48.9 |
| | 69.5 | 24.1 | 37.4 | 51.4 | 53.2 | 71.0 | 54.9 | 26.1 | 70.0 | 21.8 | 53.0 | 48.7 |
| GITHUB (CODE) | 67.0 | 1.2 | 37.3 | 51.1 | 54.1 | 71.0 | 57.3 | 27.1 | 70.1 | 21.3 | 41.2 | 47.8 |
| | 56.7 | 0.3 | 37.1 | 49.9 | 54.9 | 68.0 | 56.1 | 26.4 | 69.1 | 21.4 | 48.4 | 47.9 |
| | 62.0 | 0.2 | 37.2 | 50.2 | 54.2 | 68.0 | 56.6 | 25.8 | 70.5 | 21.8 | 54.4 | 48.7 |
| | 60.4 | 1.1 | 37.5 | 59.7 | 54.7 | 68.0 | 55.9 | 25.4 | 70.1 | 21.9 | 53.8 | 49.7 |
| | 73.6 | 0.0 | 37.3 | 55.9 | 54.1 | 71.0 | 55.4 | 25.4 | 69.9 | 21.2 | 51.4 | 49.1 |
| GITHUB (LICENSE) | 87.5 | 0.2 | 37.5 | 57.4 | 54.5 | 68.0 | 56.8 | 26.4 | 70.1 | 21.8 | 53.8 | 49.6 |
| | 74.3 | 0.0 | 37.3 | 48.9 | 54.1 | 70.0 | 57.1 | 27.1 | 70.7 | 21.7 | 48.4 | 48.4 |
| | 70.7 | 0.0 | 36.4 | 40.6 | 53.1 | 70.0 | 55.2 | 25.4 | 70.2 | 21.8 | 49.0 | 46.9 |
| | 74.8 | 0.0 | 37.3 | 60.3 | 54.8 | 69.0 | 55.9 | 27.1 | 70.0 | 21.5 | 55.6 | 50.2 |
| | 71.8 | 0.0 | 37.0 | 52.6 | 54.3 | 68.0 | 56.8 | 26.1 | 69.5 | 22.0 | 52.2 | 48.7 |
| ENRON EMAILS | 81.6 | 0.0 | 36.4 | 59.8 | 55.2 | 69.0 | 53.6 | 27.5 | 69.0 | 21.9 | 54.8 | 49.7 |
| | 70.3 | 0.0 | 37.2 | 54.9 | 54.5 | 68.0 | 57.5 | 25.4 | 70.1 | 22.4 | 51.8 | 49.1 |
| | 74.2 | 0.0 | 37.1 | 56.3 | 55.0 | 68.0 | 55.6 | 25.1 | 69.8 | 21.6 | 54.2 | 49.2 |
| | 83.9 | 0.0 | 36.7 | 55.2 | 54.8 | 69.0 | 55.9 | 25.4 | 70.4 | 21.7 | 52.2 | 49.0 |
| | 76.8 | 0.0 | 36.9 | 60.0 | 54.6 | 68.0 | 56.4 | 28.1 | 69.9 | 21.5 | 52.4 | 49.7 |
| BOOKS3 | 59.7 | 0.0 | 36.2 | 39.4 | 53.9 | 72.0 | 55.2 | 24.4 | 69.9 | 21.9 | 50.0 | 47.0 |
| | 65.4 | 0.0 | 35.9 | 65.2 | 55.7 | 67.0 | 53.3 | 25.1 | 69.9 | 21.6 | 55.8 | 49.9 |
| | 71.7 | 0.0 | 37.1 | 47.4 | 54.6 | 74.0 | 57.0 | 26.8 | 69.8 | 21.7 | 44.2 | 48.1 |
| | 74.7 | 0.0 | 36.4 | 40.7 | 53.4 | 70.0 | 55.7 | 25.4 | 69.6 | 21.6 | 41.2 | 46.0 |
| | 79.5 | 0.0 | 36.7 | 54.9 | 53.6 | 71.0 | 56.6 | 25.8 | 70.2 | 21.8 | 46.0 | 48.5 |
| PILE CC | 74.9 | 0.0 | 35.3 | 30.7 | 53.0 | 68.0 | 55.2 | 26.4 | 69.9 | 22.1 | 50.4 | 45.7 |
| | 68.0 | 0.0 | 36.3 | 45.9 | 53.4 | 72.0 | 55.6 | 27.1 | 69.6 | 21.7 | 51.4 | 48.1 |
| | 71.6 | 0.0 | 36.3 | 48.9 | 52.9 | 70.0 | 55.9 | 26.4 | 70.2 | 21.9 | 51.8 | 48.3 |
| | 57.8 | 0.0 | 34.0 | 66.3 | 55.7 | 69.0 | 49.9 | 26.1 | 69.0 | 21.4 | 57.4 | 49.9 |
| | 66.6 | 0.0 | 36.4 | 37.7 | 54.0 | 73.0 | 54.5 | 28.1 | 69.9 | 22.1 | 49.2 | 47.2 |
| USPTO BACKGROUNDS | 53.7 | 0.0 | 30.7 | 48.4 | 53.4 | 68.0 | 39.0 | 22.0 | 64.2 | 20.7 | 55.2 | 44.6 |
| | 56.7 | 0.0 | 31.0 | 19.4 | 50.6 | 69.0 | 36.9 | 24.1 | 63.3 | 21.2 | 33.4 | 38.8 |
| | 64.9 | 0.0 | 36.0 | 51.4 | 54.1 | 68.0 | 50.8 | 24.4 | 70.0 | 22.1 | 56.6 | 48.2 |
| | 54.6 | 0.0 | 35.5 | 57.2 | 55.1 | 65.0 | 52.0 | 23.7 | 68.9 | 22.0 | 56.2 | 48.4 |
| | 67.2 | 0.0 | 35.3 | 47.4 | 54.3 | 65.0 | 50.8 | 25.8 | 68.4 | 21.7 | 50.2 | 46.5 |
| PUBMED CENTRAL | 73.8 | 0.0 | 35.7 | 39.0 | 53.5 | 69.0 | 55.6 | 25.1 | 69.6 | 21.9 | 44.2 | 46.0 |
| | 75.1 | 0.0 | 36.1 | 36.3 | 53.2 | 69.0 | 54.1 | 25.1 | 69.8 | 22.6 | 44.4 | 45.6 |
| | 67.4 | 0.0 | 37.0 | 47.5 | 54.0 | 71.0 | 56.3 | 24.4 | 69.9 | 21.1 | 48.4 | 47.7 |
| | 71.1 | 0.0 | 37.2 | 55.3 | 55.6 | 68.0 | 57.0 | 24.7 | 70.0 | 22.0 | 51.0 | 49.0 |
| | 71.9 | 0.0 | 36.8 | 44.4 | 54.1 | 71.0 | 55.0 | 24.7 | 70.6 | 22.1 | 43.8 | 46.9 |

Table 6: All of the individual runs for $s = 32$ for the dialogue tasks in the Main Results.

| Model ($s$) | # Params | $EL_{10}$ (%)↓ | MA (%)↓ | WoW (F1) | ED (F1) | BST (F1) | WoI (F1) | Avg. (F1) | Epoch |
|---|---|---|---|---|---|---|---|---|---|
| NEO | 125M | 30.9 | 77.4 | **8.4** | **8.4** | **9.6** | **11.2** | **9.4** | - |
| Δ | - | - | - | +0.0 | +0.0 | +0.0 | +0.0 | +0.0 | - |
| NEO + UL$^+$ ($s = 32$) | 125M | 0.3 | 28.4 | 1.6 | 1.8 | 0.9 | 1.8 | 1.5 | 20.0 |
| | 125M | 0.8 | 27.1 | 0.1 | 0.1 | 0.0 | 0.0 | 0.0 | 18.0 |
| | 125M | 0.2 | 24.1 | 6.9 | 6.7 | 7.0 | 7.9 | 7.1 | 13.0 |
| | 125M | 3.0 | 28.7 | 2.1 | 2.5 | 1.4 | 2.3 | 2.1 | 20.0 |
| | 125M | 0.7 | 28.5 | 2.0 | 3.5 | 1.3 | 2.2 | 2.2 | 15.0 |
| NEO | 1.3B | 67.6 | 92.2 | 9.6 | **10.5** | **12.2** | **13.7** | **11.5** | - |
| Δ | - | - | - | +2.3 | +0.0 | +0.0 | +0.0 | +0.0 | - |
| NEO + UL$^+$ ($s = 32$) | 1.3B | 0.7 | 33.0 | 10.0 | 8.4 | 9.3 | 10.9 | 9.6 | 13.0 |
| | 1.3B | 1.7 | 29.8 | **11.9** | 8.4 | 10.6 | 12.4 | 10.8 | 14.0 |
| | 1.3B | 0.7 | 28.4 | 10.0 | 8.3 | 9.5 | 10.8 | 9.6 | 13.0 |
| | 1.3B | 4.2 | 31.2 | 6.4 | 5 | 4.9 | 6.8 | 5.8 | 14.0 |
| | 1.3B | 2.1 | 29.5 | 6.9 | 5.9 | 5.9 | 7.5 | 6.5 | 15.0 |
| NEO | 2.7B | 70.4 | 93.4 | 9.2 | 10.9 | **12.4** | 13.6 | 11.5 | - |
| Δ | - | - | - | +3.8 | +1.8 | +0.0 | +0.5 | +1.5 | - |
| NEO + UL$^+$ ($s = 32$) | 2.7B | 0.6 | 31.7 | 10.8 | 8.6 | 9.6 | 11.1 | 10.1 | 11.0 |
| | 2.7B | 1.1 | 32.4 | 11.9 | 9.7 | 11.5 | 12.1 | 11.3 | 10.0 |
| | 2.7B | 1.2 | 29.0 | 12.4 | 10.5 | 12.0 | 13.3 | 12.1 | 11.0 |
| | 2.7B | 3.4 | 29.9 | 8.8 | 8.2 | 8.4 | 10.3 | 8.9 | 11.0 |
| | 2.7B | 1.9 | 31.9 | **13.0** | **12.7** | **12.4** | **14.1** | **13.0** | 11.0 |

Table 7: Measuring perplexity on Pile and Wikitext corpora for the main unlearning experiments (Table 2).

| Model | # Params | Pile (PPL) ↓ | Wikitext (PPL) ↓ |
|---|---|---|---|
| NEO | 125M | 17.83 | 38.27 |
| NEO + UL | 125M | 34.02 | 75.24 |
| NEO + UL$^+$ | 125M | 577.56 | 1986.07 |
| OPT | 125M | 32.26 | 38.74 |
| NEO | 1.3B | 11.46 | 18.63 |
| NEO + UL | 1.3B | 15.56 | 20.26 |
| NEO + UL$^+$ | 1.3B | 15.83 | 26.82 |
| OPT | 1.3B | 19.55 | 19.39 |
| NEO | 2.7B | 10.44 | 16.15 |
| NEO + UL | 2.7B | 11.32 | 16.84 |
| NEO + UL$^+$ | 2.7B | 17.93 | 21.13 |
| OPT | 2.7B | 17.81 | 16.81 |

Table 8: Training compute comparison of methods mitigating privacy risks in LMs for sizes 125M, 1.3B, and 2.7B measured via FLOPs.

| Method (Size) | FLOPs |
|---|---|
| DEDUPLICATION (125M) | 2.25E+20 |
| UNLEARNING (125M) | 5.28E+13 |
| DEDUPLICATION (1.3B) | 2.34E+21 |
| UNLEARNING (1.3B) | 6.69E+14 |
| DEDUPLICATION (2.7B) | 4.86E+21 |
| UNLEARNING (2.7B) | 1.12E+15 |

Table 9: Examples from each of the 8 domains from the Pile corpora.

| Domain | Text |
|---|---|
| FREELAW | U. S. (2010) 1 Opinion of the Court NOTICE: This opinion is subject to formal revision before publication in the preliminary print of the United States Reports. Readers are requested to notify the Reporter of Decisions, Supreme Court of the United States, Washington, D. C. 20543, of any typographical or other formal errors, in order that corrections may be made before the preliminary print goes to press. SUPREME COURT OF THE UNITED STATES |
| GITHUB (CODE) | = pc func (iov *Iovec) SetLen(length int) { iov.Len = uint64(length) } func (msghdr *Msghdr) SetControllen(length int) { msghdr.Controllen = uint64(length) } func (cmsg *Cmsghdr) SetLen(length int) { cmsg.Len = uint64(length) } //sys poll(fds *PollFd, nfds int, timeout int) (n int, err error) func Poll(fds []PollFd, timeout int) (n int, err error) { if len(fds) == 0 { return poll(nil, 0, timeout) } return poll(&fds[0], len(fds), timeout) |
| GITHUB (LICENSE) | ## Permission is hereby granted, free of charge, to any person obtaining a copy # of this software and associated documentation files (the "Software"), to deal # in the Software without restriction, including without limitation the rights # to use, copy, modify, merge, publish, distribute, sublicense, and/or sell # copies of the Software, and to permit persons to whom the Software is # furnished to do so, subject to the following conditions: ## The above copyright notice and this permission notice shall be included in # all copies or substantial portions of the Software. ## THE SOFTWARE IS PROVIDED "AS IS", WITHOUT WARRANTY OF ANY KIND, EXPRESS OR # IMPLIED, INCLUDING BUT NOT LIMITED TO THE WARRANTIES OF MERCHANTABILITY, # FITNESS FOR A PARTICULAR PURPOSE |
| ENRON EMAILS | To: Hedy Govenar hgovenar@govadv.com , Mike Day MDay@GMSSR.com , Bev Hansen bhansen@lhom.com , Jeff Dasovich jdasovic@ enron.com , Susan J Mara smara@enron.com , Joseph Alamo JAlamo@enron.com , Paul Kaufman paul.kaufman@enron.com , David Parquet David.Parquet@enron.com , Rick Johnson rick.johnson@enron.com , Marcie Milner mmilner@enron.com , Sandra McCubbin Sandra.McCubbin@enron.com , Tim Belden Tim.Belden@enron.com |
| BOOKS3 | About the Publisher Australia HarperCollins Publishers (Australia) Pty. Ltd. 25 Ryde Road (PO Box 321) Pymble, NSW 2073, Australia http://www.harpercollinsebooks.com.au Canada HarperCollins Publishers Ltd. 55 Avenue Road, Suite 2900 Toronto, ON, M5R, 3L2, Canada http://www.harpercollinsebooks.ca New Zealand HarperCollins Publishers (New Zealand) Limited P.O. Box 1 Auckland, New Zealand http://www.harpercollinsebooks.co.nz United Kingdom HarperCollins Publishers Ltd. 77-85 Fulham Palace Road London, W6 8JB, UK http://www.harpercollinsebooks.co.uk |
| PILE CC | This website and its associated newspaper adheres to the Independent Press Standards Organisation's Editors' Code of Practice. If you have a complaint about editorial content which relates to inaccuracy or intrusion, then contact the Editor by clicking here. If you remain dissatisfied with the response provided then you can contact the IPSO by clicking here. Bury Free Press provides news, events and sport features from the Bury St Edmunds area. For the best up to date information relating to Bury St Edmunds and the surrounding areas visit us at Bury Free Press regularly or bookmark this page. For you to enjoy all the features of this website Bury Free Press requires permission to use cookies. Find Out More  What is a Cookie? What is a Flash Cookie? Can I opt out of receiving Cookies? |
| USPTO BACKGROUNDS | The pharmaceutical formulations of the present invention, which may conveniently be presented in unit dosage form, may be prepared according to conventional techniques well known in the pharmaceutical industry. Such techniques include the step of bringing into association the active ingredients with the pharmaceutical carrier(s) or excipient(s). In general the formulations are prepared by uniformly and intimately bringing into association the active ingredients with liquid carriers or finely divided solid carriers or both, and then, if necessary, shaping the product. The compositions of the present invention may be formulated into any of many possible dosage forms such as, but not limited to, tablets, capsules, gel capsules, liquid syrups, soft gels, suppositories, and enemas. |
| PUBMED CENTRAL | I am pleased to inform you that your manuscript has been formally accepted for publication in PLOS Computational Biology. Your manuscript is now with our production department and you will be notified of the publication date in due course. The corresponding author will soon receiving a typeset proof for review, to ensure errors have not been introduced during production. Please review the PDF proof of your manuscript carefully, as this is the last chance to correct any errors. Please note that major changes, or those which affect the scientific understanding of the work, will likely cause delays to the publication date of your manuscript. Soon after your final files are uploaded, unless you have opted out, the early version of your manuscript will be published online. The date of the early version will be your articleś publication date. |

Table 10: Examples performing extraction attacks on token sequences, showing knowledge unlearning provides protection against extraction attacks. Underlined denotes the model generated text given the prefix of length 100 as input. For the extraction attack, we utilize a naïve greedy decoding strategy.

| Domain | Status | Text |
|---|---|---|
| BOOKS3 | **Original Text** | About the Publisher Australia HarperCollins Publishers (Australia) Pty. Ltd. 25 Ryde Road (PO Box 321) Pymble, NSW 2073, Australia http://www.harpercollinsebooks.com.au Canada HarperCollins Publishers Ltd. 55 Avenue Road, Suite 2900 Toronto, ON, M5R, 3L2, Canada http://www.harpercollinsebooks.ca New Zealand HarperCollins Publishers (New Zealand) Limited P.O. Box 1 Auckland, New Zealand http://www.harpercollinsebooks.co.nz United Kingdom HarperCollins Publishers Ltd. 77-85 Fulham Palace Road London, W6 8JB, UK http://www.harpercollinsebooks.co.uk |
| | **Before Unlearning** | About the Publisher Australia HarperCollins Publishers (Australia) Pty. Ltd. 25 Ryde Road (PO Box 321) Pymble, NSW 2073, Australia http://www.harpercollinsebooks.com.au Canada HarperCollins Publishers Ltd. 55 Avenue Road, Suite 2900 Toronto, ON, M5R, 3L2, Canada http://www.harpercollinsebooks.ca New Zealand HarperCollins Publishers (New Zealand) Limited P.O. Box 1 Auckland, New Zealand http://www.harpercollinsebooks.co.nz United Kingdom HarperCollins Publishers Ltd. 77-85 Fulham Palace Road London, W6 8JB, UK http://www.harpercollinsebooks.co.uk |
| | **After Unlearning** | About the Publisher Australia HarperCollins Publishers (Australia) Pty. Ltd. 25 Ryde Road (PO Box 321) Pymble, NSW 2073, Australia http://www.harpercollinsebooks.com.au Canada HarperCollins Publishers Ltd. 55 Avenue Road, Suite 2900 Toronto, ON, M5R, 3L2, Canada http://www.harpercollins.com.au/Publishers/ Publisher: level three Level two is levels one and two together. The new face of a already great title! Level one: Just right. Level two: Great. Level three: Awesome. The BloomsburyPublishersPublishers.com.au/PublishersPublishers Levels are for bibliographic information or advanced level. s |
| PILE CC | **Original Text** | James Gurney This daily weblog by Dinotopia creator James Gurney is for illustrators, plein-air painters, sketchers, comic artists, animators, art students, and writers. You'll find practical studio tips, insights into the making of the Dinotopia books, and first-hand reports from art schools and museums. CG Art Contact or by email:gurneyjourney (at) gmail.com Sorry, I can't give personal art advice or portfolio reviews. If you can, it's best to ask art questions in the blog comments. Permissions All images and text are copyright 2015 James Gurney and/or their respective owners. Dinotopia is a registered trademark of James Gurney. For use of text or images in traditional print media or for any commercial licensing rights, please email me for permission. However, you can quote images or text without |
| | **Before Unlearning** | James Gurney This daily weblog by Dinotopia creator James Gurney is for illustrators, plein-air painters, sketchers, comic artists, animators, art students, and writers. You'll find practical studio tips, insights into the making of the Dinotopia books, and first-hand reports from art schools and museums. CG Art Contact or by email:gurneyjourney (at) gmail.com Sorry, I can't give personal art advice or portfolio reviews. If you can, it's best to ask art questions in the blog comments. Permissions All images and text are copyright 2015 James Gurney and/or their respective owners. Dinotopia is a registered trademark of James Gurney. For use of text or images in traditional print media or for any commercial licensing rights, please email me for permission. However, you can quote images or text without |
| | **After Unlearning** | James Gurney This daily weblog by Dinotopia creator James Gurney is for illustrators, plein-air painters, sketchers, comic artists, animators, art students, and writers. You'll find practical studio tips, insights into the making of the Dinotopia books, and first-hand reports from art schools and museums. CG Art Contact or by email:gurneyjourney (at) gmail.com I've been working on a CG art project for a while now, and I've been working on it for a while now. I've been working on it for a while now, and I've been working on it for a while now. I've been working on it for a while now, and I've been working on it for a while now. I've been working on a CG art project for a while now, and I've been working on it for a while |
| ENRON EMAILS | **Original Text** | Rick Shapiro rshapiro@enron.com , Jim Steffes james.d.steffes@enron.com , Alan Comnes acomnes@enron.com , Chris Calger ccalger@enron.com , Mary Hain mary.hain@enron.com , Joe Hartsoe Joe.Hartsoe@enron.com , Donna Fulton Donna.Fulton@enron.com , Steven Kean Steven.J.Kean@ enron.com , Karen Denne kdenne@enron.com , Beverly Aden beverly.aden@enron.com , Bill Votaw bill.votaw@enron.com , Carol Moffett carol. moffett@enron.com , Debora Whitehead deb |
| | **Before Unlearning** | Rick Shapiro rshapiro@enron.com , Jim Steffes james.d.steffes@enron.com , Alan Comnes acomnes@enron.com , Chris Calger ccalger@enron.com , Mary Hain mary.hain@enron.com , Joe Hartsoe Joe.Hartsoe@enron.com , Donna Fulton Donna.Fulton@enron.com , Steven Kean Steven.J.Kean@ enron.com , Karen Denne kdenne@enron.com , Beverly Aden beverly.aden@enron.com , Bill Votaw bill.votaw@enron.com , Carol Moffett carol. moffett@enron.com , Debora Whitehead |
| | **After Unlearning** | Rick Shapiro rshapiro@enron.com , Jim Steffes james.d.steffes@enron.com , Alan Comnes acomnes@enron.com , Chris Calger ccalger@enron.com , Mary Hain mary.hain@enron.com , Joe Hartsoe Joe.Hartsoe@enron.com , Donna Fulton Dabat, state+[D@calenergy.com] |
| PILE CC | **Original Text** | ? About Me Alvin McEwen is 46-year-old African-American gay man who resides in Columbia, SC. McEwen's blog, Holy Bullies and Headless Monsters, and writings have been mentioned by Americablog.com, Goodasyou.org, People for the American Way, PageOneQ.com, The Washington Post, Raw Story, The Advocate, Media Matters for America, Crooksandliars.com, Thinkprogress.org, Andrew Sullivan's Daily Dish, Melissa Harris-Perry, The Last Word with Lawrence O'Donnell, Newsweek, The Daily Beast, The Washington Blade, and Foxnews.com. In addition, he is also a past contributor to Pam's House Blend,Justice For All, LGBTQ Nation, and Alternet.org. He is a present contributor to the Daily Kos and the Huffington Post, He is the 2007 recipient of the Harriet Daniels Hancock Volunteer of the Year Award and |
| | **Before Unlearning** | ? About Me Alvin McEwen is 46-year-old African-American gay man who resides in Columbia, SC. McEwen's blog, Holy Bullies and Headless Monsters, and writings have been mentioned by Americablog.com, Goodasyou.org, People for the American Way, PageOneQ.com, The Washington Post, Raw Story, The Advocate, Media Matters for America, Crooksandliars.com, Thinkprogress.org, Andrew Sullivan's Daily Dish, Melissa Harris-Perry, The Last Word with Lawrence O'Donnell, Newsweek, The Daily Beast, The Washington Blade, and Foxnews.com. In addition, he is also a past contributor to Pam's House Blend,Justice For All, LGBTQ Nation, and Alternet.org. He is a present contributor to the Daily Kos and the Huffington Post, He is the 2007 recipient of the Harriet Daniels Hancock Volunteer of the Year Award and |
| | **After Unlearning** | ? About Me Alvin McEwen is 46-year-old African-American gay man who resides in Columbia, SC. McEwen's blog, Holy Bullies and Headless Monsters, and writings have been mentioned by Americablog.com, Goodasyou.org, People for the American Way, PageOneQ.com, The Washington Post, Raw Story, The Advocate, Media Matters for America, Crooksandliars.com, Thinkprogress, and more. The British singer has been in the news for his recent singles, including "I'm Not Sure" and "What Makes You Beautiful." The singer has been in the news for his recent singles, including "I'm Not Sure" and "What Makes You Beautiful." The singer has been in the news for his recent singles, including "I'm Not Sure" |

Table 11: Forgetting Threshold for GPT-NEO LMs for varying $n$.

| Model (Size) | $EL_5(\%)$ Threshold | $EL_{10}(\%)$ Threshold | $EL_{20}(\%)$ Threshold | $EL_{40}(\%)$ Threshold | MA(%) Threshold |
|---|---|---|---|---|---|
| GPT-NEO (1.3B) | 7.85 | 5.68 | 4.07 | 2.66 | 33.27 |

Table 12: The average of the 9 classification tasks for GPT-NEO + UL$^+$ for the 1.3B LM when performing unlearning until the Forgetting Threshold for each $n$.

| Model (Size) | LM Avg. (Acc) |
|---|---|
| $\textbf{EL}_5$ | 49.93 |
| $\textbf{EL}_{10}$ | 49.93 |
| $\textbf{EL}_{20}$ | 49.85 |
| $\textbf{EL}_{40}$ | 49.88 |

