# OpenReview forum: "Knowledge Unlearning for Mitigating Privacy Risks in Language Models"
_ICLR.cc/2023/Conference — Submitted to ICLR 2023_

### Official Review · Reviewer_9RUh · 2022-10-15

**Confidence:** 4
**Correctness:** 3
**Technical Novelty And Significance:** 3
**Empirical Novelty And Significance:** 3
**Recommendation:** 6

**Clarity, Quality, Novelty And Reproducibility:**

The work appears novel, the writing is clear and is easy to read, and the approach is simple and straightforward enough that reproducing the results is likely feasible.

**Strength And Weaknesses:**

Strengths
---
Unlearning for large LMs is certainly a timely problem since large LMs have exploded in popularity, and are used as the foundation for almost all downstream NLP tasks, exacerbating any privacy concerns inherent in the pretrained LM. Thus, this work tackles an important problem of helping to mitigate some of the privacy issues that are likely to arise through the widespread use of large pretrained LMs.

The proposed approach is very simple yet effective and extremely efficient in comparison to retraining from scratch, which is generally intractable for large pretrained LMs. This makes the proposed approach very practical and likely to be used in the real world; it is also a much more appealing option than data preprocessing and expensive differential privacy methods.

Experiments using various LM model sizes, and evaluations on 9 different domain datasets provide decent evidence the proposed approach is a potentially viable unlearning method that warrants further study.

Weaknesses
---
The forgetting guarantee requires extra validation data.

No results that vary the size of n, the hyperparameter used for the EL metric. This can give better insight into how much of the target examples are unlearned when the prompt varies in size. Additionally, how much does the success of the EL metric vary depending on which n tokens are used as a prompt for this metric?

Only a small number of examples (32) are randomly selected to be unlearned. Have the authors tried unlearning much larger portions of the training data and observing the effect on the resulting model?

It is still not quite clear to me why unlearning a larger batch of examples (128 vs 32) performs significantly worse than sequentially unlearning smaller batches of 32 examples at a time.

Questions/Comments
---
How is the "hardness" of forgetting defined. Is it how much predictive performance changes when unlearning an example? Or, is it defined as the number of epochs to successfully achieve forgetting for the given input?

The improved forgetting and utility performance of the larger GPT-NEO model compared to the smaller versions is interesting. I would think these results suggest unlearning is "easier" for larger LMs with potentially more representational power.

Do the authors have any insight into the relation between a model that uses the proposed unlearning mechanism and a model retrained from scratch without the target training examples to be removed?

Minor Weaknesses
---
Table 3 is not color-blind friendly, consider showing prefixes as underlined text instead of blue-colored text.

Footnote 6 (in the conclusion) should come after the period.

**Summary Of The Paper:**

The authors propose a simple approach for unlearning specific token sequences in large pretrained language models (LMs). To unlearn a specific sequence of tokens x, the proposed approach simply negates the original training objective of minimizing the negative log-likelihood of x; this procedure is also known as unlikelihood training. To measure the effectiveness of unlearning, the authors introduce two metrics, extraction likelihood (EL), and memorization accuracy (MA). EL is measured as the average n-gram overlap between the output of the LM given the tokens in x up to token t and the ground-truth tokens greater than or equal to t. MA is the fraction of tokens output from the LM that match the tokens in x. The "forgetting" of x is achieved when the EL and MA of x is lower than the average EL and MA over token sequences in a validation set not seen during training.

Experiments are performed using three different sizes of the GPT-NEO LM pretrained on all the PILE corpora. After unlearning, the LM is evaluated using 9 different domain datasets spanning a wide range of downstream NLP tasks. The results show the proposed unlearning approach achieves better unlearning (evaluated using the proposed metrics described above) than the most relevant competing method (a data pre-processing de-duplication approach) while maintaining better utility on most the of downstream NLP tasks; this is especially evident as the size of the GPT-NEO model increases. The authors also find their proposed approach is up to 3,500,000x more efficient than retraining from scratch.

**Summary Of The Review:**

Given the ubiquity of large pretrained LMs in NLP today, concerns related to privacy are likely to arise, thus new proposals for tackling these privacy implications is important. This work presents a simple, efficient, and effective approach for targeted unlearning in large LMs, and is likely worthy of further research.

Post-Rebuttal
---
This work applies gradient ascent to a new domain, namely large language models that are typically prohibitively expensive to retrain, enabling an efficient way to remove specific training sequences post-hoc. However, the empirical evaluation does not seem to fully support the main claims made in this paper. A more thorough investigation analyzing the predictive performance of the LLM as more instances are unlearned would greatly improve this paper and ultimately help readers and practitioners better understand the limitations of this approach. Also, better understanding the difference between sequential and batch unlearning would also significantly improve this paper.

---

> ### Author Response · Authors · 2022-11-15
> **Response to Reviewer 9RUH**
>
> Hello Reviewer 9RUh,
>
> Thank you for your positive review. We address your concerns and questions below.
>
> > Weakness #1: The forgetting guarantee requires extra validation data.
>
> This is true. Getting the threshold value for EL & MA does require extra validation data that is of similar distribution with the training corpora.
>
> > Weakness #2: No results that vary the size of n, the hyperparameter used for the EL metric. This can give better insight into how much of the target examples are unlearned when the prompt varies in size. Additionally, how much does the success of the EL metric vary depending on which n tokens are used as a prompt for this metric?
>
> Thank you for your comment. We additionally provide varying the size of n (how many consecutive tokens have to be successfully extracted in order for the attack to be considered successful) of values [5, 10, 20, 40] and how it affects the EL value for the GPT-Neo 1.3B LM in the general response section. We have included the ablation results in the appendix of the revised version.
>
> > Weakness #3: Only a small number of examples (32) are randomly selected to be unlearned. Have the authors tried unlearning much larger portions of the training data and observing the effect on the resulting model?
>
> We observed that the LM performance deteriorates when trying to unlearn 128 samples at once. We also tried unlearning 512 samples, but it resulted in much more significant deterioration. However, positive results were shown by dividing the samples into chunks and sequentially unlearning them. So we do not further report the numbers for >128 in the paper.
>
> > Weakness #4: It is still not quite clear to me why unlearning a larger batch of examples (128 vs 32) performs significantly worse than sequentially unlearning smaller batches of 32 examples at a time.
>
> As replied to Reviewer 4t9e as well, our intuition is that when trying to make gradient updates on multiple samples at once, it will result in optimizing towards multiple directions on the loss landscape. Instead, optimizing towards a narrower direction at a time (sequential unlearning) proves to result in better forgetting without the associated performance degradation. However, the authors agree that further analysis can be made in future work where we can attempt to scale the total # of samples being forgotten and see how the LM performs.
>
> > Question #1: How is the "hardness" of forgetting defined. Is it how much predictive performance changes when unlearning an example? Or, is it defined as the number of epochs to successfully achieve forgetting for the given input?
>
> In our work, we refer to the “hardness” of forgetting as a combination of both the number of epochs needed to achieve successful forgetting and the associated performance degradation that results from the forgetting. Future work might attempt to divide the two criteria to provide a more detailed analysis.
>
> > Comment #1: The improved forgetting and utility performance of the larger GPT-NEO model compared to the smaller versions is interesting. I would think these results suggest unlearning is "easier" for larger LMs with potentially more representational power.
>
> The authors agree that the reason larger LMs perform unlearning “easier” is potentially because of stronger representational power. It would be interesting to test this phenomenon for much larger LMs with ten billion parameter sizes to see if the phenomenon is scalable.
>
> > Question #2: Do the authors have any insight into the relation between a model that uses the proposed unlearning mechanism and a model retrained from scratch without the target training examples to be removed?
>
> It would be interesting to test the LM retrained from scratch without the target training examples. But as you mentioned, retraining large pretrained LMs from scratch is generally intractable. However, the authors observed that LMs do show a high EL and MA for target token sequences from the validation corpora that were not seen during training, but are very similar to the token sequences that are extractable. This is especially the case for ‘structured’ token sequences such as code data which have specific patterns that may be repeating. In this scenario, the authors think the difference will not be very significant. On the other hand, further exploring the ‘generalization’ capability of unlearning is an interesting future work.
>
> > Minor Weaknesses: Table 3 is not color-blind friendly, consider showing prefixes as underlined text instead of blue-colored text. Footnote 6 (in the conclusion) should come after the period.
>
> Thank you for your comments. We have changed Table 3 to be color friendly by underlining the text and fixed the Footnote 6 typo.

---

> > ### Comment · Reviewer_9RUh · 2022-11-16
> > **Response**
> >
> > I thank the authors for their response clarifying my concerns, and I have increased my score accordingly. Although there exist some concern over the limited novelty of the proposed method (gradient ascent), the application of gradient ascent for the specific task of machine unlearning in large LLMs seems entirely novel and is very timely as large LLMs are pervasive in deep learning research today and have many potential commercial applications. I also think it is non trivial to apply efficient and accurate unlearning techniques to large LLMs whose outputs contain arbitrary-length token sequences, which is significantly different from models designed for image classification, for example. Overall, this approach is simple, appears effective, is likely to actually be used in practice, and is also likely to prompt further research into machine unlearning for large LLMs.

---

### Official Review · Reviewer_jyEc · 2022-10-25

**Confidence:** 4
**Correctness:** 3
**Technical Novelty And Significance:** 2
**Empirical Novelty And Significance:** 3
**Recommendation:** 6

**Clarity, Quality, Novelty And Reproducibility:**

The paper is well-written and well-structured. The proposed method seems novel to me and the paper seems reproducible. However it is lacking justification for the choice of baselines and metrics.

**Strength And Weaknesses:**

Strengths:

1. The method is simple and intuitive and has nearly no overhead.
2. The problem is timely and relevant.


Weaknesses:

1. There are some experimental and design decisions that seem arbitrary and I don't really see the reasoning behind:
    a. The choice of baseline: I don't really see why deduplication is chosen as a baseline to compare to? It seems the most irrelevant and also the most compute-intensive one. The proposed method is not at all similar to deduplication, which is a pre-processing method. Also, deduplication itself is really not a valid 'privacy protection method. I think the most appropriate baseline would actually be doing differential privacy, like DP-SGD, and then doing a comparison of memorization. So for instance if you wanna delete sample `a', you'd train a model on the full dataset, then unlearn it using your method, and then measure `a's memorization before and after. Then, you'd compare this with memorization of 'a' but under a model trained with DPSGD. DPSGD is the best point of comparison as by design, it is supposed to protect the membership, so it is like a pro-active approximate deletion method.

b. The proposal of the privacy metrics: I wonder why the authors made up their own metrics, and also came up with an arbitrary privacy guarantee which is based on their two metrics. Why not just use membership inference attack recall [1,2] and exposure metric [3], which are commonly used and established metrics? These two basically do what the currently proposed metrics do.

c. apart from not having an appropriate baseline, the paper also fails to even qualitatively compare, or even introduce in related work, to 'approximate deletion' methods and other descent-based methods such as [4,5] which are also aimed at deletion, w/o full re-training.


2. Figure 1 is actually inaccurate in terms of DP, as you do not need to re-train with DP every time you get a deletion request, given how DP is actually defined.

[1]   Shokri, Reza, et al. "Membership inference attacks against machine learning models." 2017 IEEE symposium on security and privacy (SP). IEEE, 2017.

[2] Mireshghallah, Fatemehsadat, et al. "Quantifying privacy risks of masked language models using membership inference attacks." arXiv preprint arXiv:2203.03929 (2022).

[3] Carlini, Nicholas, et al. "The secret sharer: Evaluating and testing unintended memorization in neural networks." 28th USENIX Security Symposium (USENIX Security 19). 2019.

[4] Izzo, Zachary, et al. "Approximate Data Deletion from Machine Learning Models." arXiv preprint arXiv:2002.10077 (2020).

[5] Neel, Seth, Aaron Roth, and Saeed Sharifi-Malvajerdi. "Descent-to-delete: Gradient-based methods for machine unlearning." Algorithmic Learning Theory. PMLR, 2021.

**Summary Of The Paper:**

This paper proposes a simple method for unlearning training samples, to comply with GDPR (and other privacy acts') right-to-be-forgotten statement, which gives each person the right to delete their data at any time they want. The proposed unlearning method for language models involves doing a gradient ascent (instead of the usual descent) step, minimizing the likelihood of the training samples. The authors empirically show that this negative step can decease the memorization of the sample, based on the metrics they introduce themselves. They also compare the unlearning method they introduce to a deduplication baseline.  They also demonstrate that unlearning samples sequentially provides a better utility, than unlearning samples all at once.

**Summary Of The Review:**

The paper's idea is interesting, simple, and effective. However, I think a better baseline and better metrics are needed to be able to actually say that it outperforms other methods. I think a comparison with DP is needed, and also a comparison with other approximate deletion methods is also needed.

---

> ### Author Response · Authors · 2022-11-15
> **Response to Reviewer jyEC**
>
> Hello Reviewer jyEC,
>
> Thank you for taking the time to provide us with valuable feedback. We address your questions/concerns in the response below.
>
> > a. The choice of baseline: I don't really see why deduplication is chosen as a baseline to compare to? It seems the most irrelevant and also the most compute-intensive one. The proposed method is not at all similar to deduplication, which is a pre-processing method. Also, deduplication itself is really not a valid 'privacy protection method. I think the most appropriate baseline would actually be doing differential privacy, like DP-SGD, and then doing a comparison of memorization. So for instance if you wanna delete sample a', you'd train a model on the full dataset, then unlearn it using your method, and then measure a's memorization before and after. Then, you'd compare this with memorization of 'a' but under a model trained with DPSGD. DPSGD is the best point of comparison as by design, it is supposed to protect the membership, so it is like a pro-active approximate deletion method.
>
> To address this concern, we have added a DP Decoding method for LLMs [1] as additional baseline approaches that can be applied w/o full re-training. The authors agree that an LM pretrained from scratch using DP-SGD is a good comparison to have [2], but as mentioned in the limitation section of the initial draft, it was computationally infeasible to implement this approach since it requires pretraining an LM with billion parameters from scratch which requires training the LM for thousands of GPU days. Moreover, applying DP-SGD to an already pretrained LM is not suitable for our set-up, as explained in detail in the general response section.
>
> > b. The proposal of the privacy metrics: I wonder why the authors made up their own metrics, and also came up with an arbitrary privacy guarantee which is based on their two metrics. Why not just use membership inference attack recall [1,2] and exposure metric [3], which are commonly used and established metrics? These two basically do what the currently proposed metrics do.
>
> Regarding the comment about the evaluation metric, we want to clarify that we did not come up with Memorization Accuracy (MA), but was first proposed by [3] that quantifies how much the LM has memorized the given target token sequences. We also want to clarify that the objective of our approach is not to provide general protection (e.g. membership inference attacks), but only for the given target token sequences. Also, the exposure metric is not an appropriate metric since it is limited to quantifying the risk of extracting previously defined ‘canaries’, which in previous literature [4], are limited to quantifying extraction of one or three words. Since the exposure metric requires sampling all the possible outcomes, trying to use it to quantify the risks of a 200-token sequence is infeasible since the randomness space will grow exponentially to the target token sequence length. We have clarified why we do not include the previous metric in our revised draft.
>
> > c. apart from not having an appropriate baseline, the paper also fails to even qualitatively compare, or even introduce in related work, to 'approximate deletion' methods and other descent-based methods such as [4,5] which are also aimed at deletion, w/o full re-training.
>
> We thank the reviewers for the related works. We will make sure to include them in the related works section. However, we want to point out that [4,5] are both descent-based methods for linear and logistic models and are not applicable for large neural models that are of billion parameters. We instead include other comparable baseline approaches.
>
> > Figure 1 is actually inaccurate in terms of DP, as you do not need to re-train with DP every time you get a deletion request, given how DP is actually defined.
>
> Thanks for the comment. Given the way DP is defined, there is no straightforward approach to responding to a deletion request utilizing traditional DP algorithms such as DP-SGD. DP Decoding [1] aims to overcome this shortcoming by proposing a decoding strategy that can guarantee DP, but with significant performance loss, as shown in the general response section. We have clarified in Figure 1 that we are referring to traditional DP algorithms that require parameter updates, such as DP-SGD, and not referring to it broadly as DP since it might be misleading.

---

> > ### Author Response · Authors · 2022-11-15
> > **(Continued Response)**
> >
> > **References**
> >
> > [1] Majmudar, Jimit, Christophe Dupuy, Charith Peris, Sami Smaili, Rahul Gupta, and Richard Zemel. "Differentially Private Decoding in Large Language Models." arXiv preprint arXiv:2205.13621 (2022).
> >
> > [2] Anil, Rohan, Badih Ghazi, Vineet Gupta, Ravi Kumar, and Pasin Manurangsi. "Large-scale differentially private bert." arXiv preprint arXiv:2108.01624 (2021).
> >
> > [3] Kushal Tirumala, Aram H Markosyan, Luke Zettlemoyer, and Armen Aghajanyan. Memorization without overfitting: Analyzing the training dynamics of large language models. arXiv preprint arXiv:2205.10770, 2022.
> >
> > [4] Carlini, Nicholas, et al. "The secret sharer: Evaluating and testing unintended memorization in neural networks." 28th USENIX Security Symposium (USENIX Security 19). 2019.

---

> > > ### Comment · Reviewer_jyEc · 2022-11-17
> > > **Post rebuttal**
> > >
> > > I thank the authors for their response. The DP decoding baseline and additional discussion surrounding it are convincing to me. As such, I will increase my score. Thank you!

---

### Official Review · Reviewer_4t9e · 2022-10-26

**Confidence:** 3
**Correctness:** 3
**Technical Novelty And Significance:** 3
**Empirical Novelty And Significance:** 4
**Recommendation:** 6

**Clarity, Quality, Novelty And Reproducibility:**

While there may not be a lot of novelty in the methods designed in this work, I do believe that it has a significant contribution to the field of machine unlearning in large LMs. The quality of writing is good, and the authors provide code for reproducibility.

**Strength And Weaknesses:**

## Interesting Results and Strengths
1. The paper is very well written and follows easily, providing enough background to the reader wherever needed.
2. The experimental setup is comprehensive with LM performance compared to a variety of tasks. And benchmarks the results for future work to improve upon.
3. Many of the experimental decisions taken in this work were interesting: for example, using OPT method to replicate a deduplicated dataset, and using data from the Extraction dataset to measure unlearning.
4. While the LMs scale to larger sizes, it takes fewer epochs for the target sequences to be forgotten.
5. Sequential Unlearning is more Stable than Batch Unlearning. Do you have any intuition as to why this happens?

## Weaknesses
1. Thresholds: The thresholds should be set as mean + k* standard deviation and not just the mean. This current approach does not guarantee that with a high probability the examples are forgotten. How do the results change after making this modification?
2. EL_n value: How was the value of n set to 10
3. The Ablation/Understanding section of this paper is weak. In particular, the authors do ask all the right questions: "Why Are Some Instances Harder to Forget?" "Why Are Some Instances Harder to Forget?"  but the study only provides empirical evidence to rehash the same phenomenon without much understanding gained for the reader. The experimental study rather answers the question: "Are Some Instances Harder to Forget?" and so on.
4. Comparisons with prior work: While there is no significant literature for unlearning in the LM domain, it would have been nice for the authors to utilize the compare the best unlearning algorithms in the vision domain, and see if they can leverage the advances in the vision domain in the NLP domain.
5. The authors use the term "guarantees privacy" for model extraction attacks. I would not use terminology like guarantee for empirical evidence.

**Summary Of The Paper:**

This work performs knowledge unlearning in large language models by attempting to fine-tune a converged model for a few epochs with a negative loss corresponding to the examples to the forgotten. As compared to past work which considers data duplication as a strategy towards reducing memorization, this work shows that knowledge unlearning can be very efficient, and does not reduce the LM performance significantly.

The authors also propose a metric to quantify if an example is forgotten, and call it the EL or Extraction Likelihood. Coupled with a metric for memorization, the work

Some other interesting findings include that the domain of the data to be removed has a large impact on the ease of forgetting of the example. Further, removing small batches of data sequentially is better than removing large batches at once.

**Summary Of The Review:**

I find that this paper does an important job of benchmarking initial results for unlearning in the context of large language models. The paper presents an optimistic picture for LMs suggesting that even with baseline unlearning methods, one can achieve forgetting without hurting LM perform. The experimental study is comprehensive and allows future work to build upon with the released code.

---

> ### Author Response · Authors · 2022-11-15
> **Response to Reviewer 4t9e**
>
> Hello Reviewer 4t9e,
>
> Thank you for your comments. We address your questions/concerns in the response below.
>
> > Question #1: Sequential Unlearning is more Stable than Batch Unlearning. Do you have any intuition as to why this happens?
>
> Our hypothesis is that when trying to make gradient updates on multiple samples at once, it will result in optimizing towards multiple directions on the loss landscape. Instead, optimizing towards a narrower direction at a time by dividing the multiple samples into different chunks and performing sequential unlearning proves to result in better forgetting without the associated performance degradation. We plan to explore exactly why this happens during gradient ascent in future work.
>
> > Weakness #1: Thresholds: The thresholds should be set as mean + k* standard deviation and not just the mean. This current approach does not guarantee that with a high probability the examples are forgotten. How do the results change after making this modification?
>
> The threshold value used to empirically determine a target token sequence to be forgotten is measured by getting the average of 10,000 instances (each 200 tokens long)  from the validation corpora. Thus, we determined the sample size was big enough to simply utilize the average value as the threshold value.
>
> > Weakness #2: EL_n value: How was the value of n set to 10
>
> We show the results of GPT-Neo 1.3B as we vary the value of n among [5, 10, 20, 40] in the general response section and have also included these results in the revised appendix.
>
> > Weakness #3: The Ablation/Understanding section of this paper is weak. In particular, the authors do ask all the right questions: "Why Are Some Instances Harder to Forget?" "Why Are Some Instances Harder to Forget?" but the study only provides empirical evidence to rehash the same phenomenon without much understanding gained for the reader. The experimental study rather answers the question: "Are Some Instances Harder to Forget?" and so on.
>
> In this work, we show some empirical results that can be the starting point for answering the question “Why Are Some Instances Harder to Forget?”. For example, we show that it is easier to unlearn relatively structured instances such as code or emails while unlearning relatively unstructued instances results in more degradation of general LM performance. We have changed our tone in the draft and have explicitly mentioned that we provide a starting point of analysis of understanding exactly what happens to the LM during gradient ascent.
>
> > Weakness #4: Comparisons with prior work: While there is no significant literature for unlearning in the LM domain, it would have been nice for the authors to utilize the compare the best unlearning algorithms in the vision domain, and see if they can leverage the advances in the vision domain in the NLP domain.
>
> In the vision domain, simply performing gradient ascent results in severe deterioration of general performance since they are only concerned with classification tasks with only 10’s of classes. Thus, most recent approaches in the vision domain try to mitigate severe deterioration when performing unlearning. We show that simply applying gradient ascent results in successful unlearning while not resulting in a degradation of performance, especially as LMs scale. Regarding the lack of comparison of prior work, we included a decoding scheme during inference that can provide DP guarantees [1] as an additional baseline. The details and results are provided in the general response section.
>
> > Weakness #5: The authors use the term "guarantees privacy" for model extraction attacks. I would not use terminology like guarantee for empirical evidence.
>
> Thank you for your comment. We made sure to tone down on our usage of the word ‘guarantee’ since we only test with empirical evidence and instead use the word ‘protection’ in our revised version.
>
> **References**
>
> [1] Majmudar, Jimit, Christophe Dupuy, Charith Peris, Sami Smaili, Rahul Gupta, and Richard Zemel. "Differentially Private Decoding in Large Language Models." arXiv preprint arXiv:2205.13621 (2022).

---

> > ### Comment · Reviewer_4t9e · 2022-11-17
> > **Response to Rebuttal**
> >
> > >The threshold value used to empirically determine a target token sequence to be forgotten is measured by getting the average of 10,000 instances (each 200 tokens long) from the validation corpora. Thus, we determined the sample size was big enough to simply utilize the average value as the threshold value.
> >
> > I do not think this argument is sound. Indeed, you can estimate the mean to a very close approximation, however, at the time of forgetting you will once again be doing this at an instance level. Therefore, the probability that a sample is greater than its mean is only 0.5. mean + 3sigma is a good threshold. Please correct me if I am wrong.
> >
> >
> > Thank you for the other clarificatons.

---

> > > ### Comment · Reviewer_4t9e · 2022-12-10
> > > **Still awaiting Author Response**
> > >
> > > Dear Authors,
> > >
> > > I am waiting for a response from your end regarding my concern that I raise above.
> > >
> > > Additionally, after reading responses to other reviewers I am concerned about the numbers reported in the paper. In particular, the authors note in a different response that
> > >
> > > > We observed that the LM performance deteriorates when trying to unlearn 128 samples at once. We also tried unlearning 512 samples, but it resulted in much more significant deterioration. However, positive results were shown by dividing the samples into chunks and sequentially unlearning them. So we do not further report the numbers for >128 in the paper.
> > >
> > > However, in response to my query about testing other metrics, they say that
> > > > In the vision domain, simply performing gradient ascent results in severe deterioration of general performance since they are only concerned with classification tasks with only 10’s of classes. Thus, most recent approaches in the vision domain try to mitigate severe deterioration when performing unlearning. We show that simply applying gradient ascent results in successful unlearning while not resulting in a degradation of performance, especially as LMs scale. Regarding the lack of comparison of prior work, we included a decoding scheme during inference that can provide DP guarantees [1] as an additional baseline. The details and results are provided in the general response section.
> > >
> > > These answers are contradictory, and my concern about relevant experiments is stronger after reading the same. I would appreciate a response from your side.
> > >
> > > I would like to see a graph of the LM performance deterioration with the percentage of data unlearned, say up to 20% using different metrics to make this a comprehensive study.

---

> > > > ### Author Response · Authors · 2022-12-11
> > > > **Response to Reviewer**
> > > >
> > > > Hello Reviewer 4t9e,
> > > >
> > > > Sorry for the late response. I believe setting the threshold for forgetting is a design choice since it is still an *empirical* measure of how much of the given 200 token sequences have been memorized by the LM. I do not believe setting the threshold to +3 sigma will have a huge impact on our main experimental results.
> > > >
> > > > I believe your concern regarding the contradiction is the authors saying that (1) knowledge unlearning  (via simple gradient ascent) results in severe deterioration when trying to forget >128 samples at once (referred to as **batch** unlearning in the paper) and saying that (2) knowledge unlearning via simple gradient ascent does NOT result in a severe deterioration compared to the experimental setting in the vision domain when in (1) we say it does for >128 samples.
> > > >
> > > > As shown in Figure 2 (b) where we show forgetting of 128 samples but performing **sequential** unlearning, it results in little to no degradation, especially for the largest LM (2.7B). So both paragraphs are not contradictory since we show that gradient ascent results in little to no degradation of capabilities when performing sequential unlearning (it does result in degradation for **batch** unlearning). One interesting aspect is exploring how **sequential** unlearning helps with algorithms in the vision domain, which is out-of-scope from our work.
> > > >
> > > > Hope this clears your concern.

---

### Official Review · Reviewer_hH19 · 2022-10-31

**Confidence:** 4
**Correctness:** 4
**Technical Novelty And Significance:** 2
**Empirical Novelty And Significance:** 3
**Recommendation:** 5

**Clarity, Quality, Novelty And Reproducibility:**

* I have some concerns regarding the technical novelty. The general idea of gradient ascent on some data to cause models to avoid learning is an idea that has been well-explored in the literature. For example, the canonical Ganin et al. 2015 paper that proposes "gradient reversal" layers. More recently, there is the large body of work on machine unlearning. Although the paper is correct that this is mainly focused on CV and basic regression tasks, there isn't a ton of new technical novelty required to apply the ideas to NLP.
* I find it weird to discuss de-duplication as a sort of baseline. Deduplication should be almost always used as a first data processing stage to mitigate memorization (Kandpal et al. 2022; Carlini et al. 2022), and in that sense, it's quite orthogonal to the method proposed here.
* The quality and clarity is otherwise great, easy to read and follow. Reasonable experiments.


**Strength And Weaknesses:**

Strengths:
* Memorization is a key problem to stop, and more empirical techniques are needed that do not sacrifice performance as differential privacy does.

Weaknesses:
* One of my big concerns is regarding the privacy evaluation of the method. The evaluation focuses specifically on preventing exact verbatim memorization. A trivial baseline for stopping verbatim memorization is to just manually block the model from generating exact strings in the training set, potentially using a suffix tree like is done in Kandpal et al. 2022 and Katherine Lee et al. 2022. Of course, this trivial baseline cannot prevent models from privacy risks such as (1) generating paraphrases of training texts, (2) leaking private information in a QA or dialogue setting (e.g., ask someone's SSN), and (3) cannot stop membership inference attacks. To show the benefits of the method, it'd be great to see results in some of the above (or related) evaluations. For example, run the membership inference methods from Kandpal et al. 2022, Carlini et al. 2020, or others.
* Definitely checkout the privacy onion paper (Carlini et al. 2022) for another potential risk---unlearning some data can expose others.

**Summary Of The Paper:**

Memorization is a key emerging privacy risk in recent large-scale language models, wherein models can remember and regurgitate verbatim samples from the training set. This paper proposes to alleviate this problem via "knowledge unlearning", which is essentially running stochastic gradient *ascent* on samples that have been identified as memorized. The paper explores a few details on how this is done correctly (proposing sequential unlearning), and shows that the method can mitigate many verbatim generations.

**Summary Of The Review:**

I find the method to be quite similar to past ideas and the privacy evaluation to be a bit limited.

---

> ### Author Response · Authors · 2022-11-15
> **Response to reviewer hH19**
>
> Hello Reviewer hH19,
>
> Thank you for taking the time to provide us with valuable feedback. We respond to specific review comments below.
>
> > Weakness #1: One of my big concerns is regarding the privacy evaluation of the method. The evaluation focuses specifically on preventing exact verbatim memorization. A trivial baseline for stopping verbatim memorization is to just manually block the model from generating exact strings in the training set, potentially using a suffix tree like is done in Kandpal et al. 2022 and Katherine Lee et al. 2022. Of course, this trivial baseline cannot prevent models from privacy risks such as (1) generating paraphrases of training texts, (2) leaking private information in a QA or dialogue setting (e.g., ask someone's SSN), and (3) cannot stop membership inference attacks. To show the benefits of the method, it'd be great to see results in some of the above (or related) evaluations. For example, run the membership inference methods from Kandpal et al. 2022, Carlini et al. 2020, or others.
>
> We want to highlight that using a suffix tree has critical flaws in terms of scalability and computational overhead. For example, say there is a phone number 123-456-78910 that is part of the suffix tree. Then, the LM will not be able to generate 123 or 456, which may be required in a different context. As the size of the suffix tree grows, this side-effect might further be exacerbated. Second, the computation overhead of utilizing a suffix tree is non-trivial. While the time complexity of naive greedy decoding might be O(n) where n is the length of the max generation sequence length, utilizing a suffix tree will result in a time complexity of O(n * m * # of target token sequences) during each inference step where m represents the average length of the target token sequences that we want to prevent verbatim generation on. Because of these two critical limitations, we do not include utilizing a suffix tree as a baseline method.
>
> We describe why we do not consider membership inference methods for evaluation in the general response section. However, we do agree that our current evaluation setting is limited to evaluating only verbatim generation and quantifying the extent of targeted extraction attacks. Extending the current evaluation setting to scenarios such as quantifying private information leakage in a dialogue or QA setting or evaluating on paraphrases of the training texts is an interesting future work to further explore.
>
> > Weakness #2: Definitely checkout the privacy onion paper (Carlini et al. 2022) for another potential risk---unlearning some data can expose others.
>
> Thank you for pointing out this related work. We have revised our related work and limitations sections to include this work.
>
> > Concern #1: I have some concerns regarding the technical novelty. The general idea of gradient ascent on some data to cause models to avoid learning is an idea that has been well-explored in the literature. For example, the canonical Ganin et al. 2015 paper that proposes "gradient reversal" layers. More recently, there is the large body of work on machine unlearning. Although the paper is correct that this is mainly focused on CV and basic regression tasks, there isn't a ton of new technical novelty required to apply the ideas to NLP.
>
> While we agree that the concept of gradient ascent has existed in previous literature, we believe we made some non-trivial findings since we showed that the simple approach of gradient ascent, which brings severe degradation in the previous machine unlearning set-ups and requires other complex algorithms to mitigate the degradation, results in little to no degradation of general performance for LMs, especially as the LMs scale to larger sizes.
>
> > Concern #2: I find it weird to discuss de-duplication as a sort of baseline. Deduplication should be almost always used as a first data processing stage to mitigate memorization (Kandpal et al. 2022; Carlini et al. 2022), and in that sense, it's quite orthogonal to the method proposed here.
>
> Thank you for your comment regarding the need for more comparable baselines. We have additionally included a baseline approach that can be applied w/o full re-training [1]. The summary of results is provided in the general response section.
>
> **References**
>
> [1] Majmudar, Jimit, Christophe Dupuy, Charith Peris, Sami Smaili, Rahul Gupta, and Richard Zemel. "Differentially Private Decoding in Large Language Models." arXiv preprint arXiv:2205.13621 (2022).

---

### Author Response · Authors · 2022-11-15
**General Response (Updates on Baselines and Addressing Common Concerns)**

We thank all the reviewers for taking the time and providing us with valuable feedback. We want to provide a general response regarding clarification about the problem setting, updates regarding the new baseline, and the effect of varying the hyperparameter for our proposed EL metric.

**Clarification about problem setting**

We want to highlight some points that were unclear in our draft regarding our evaluation setting. We did not include prior evaluation settings used in the privacy literature, such as membership inference attacks, because our proposed method is not meant to provide general protection against all attacks, but only for targeted extraction attacks specifically for given token sequences. For example, member inference attacks assume that there are model-generated outputs that are either randomly generated or generated verbatim from the training corpora. The task is to classify whether the generated output is part of the membership (training corpora) or not. This tests how likely it is for an adversary to extract any portion of training data. We are, however, only concerned with a scenario where individuals practice their Right-To-Be-Forgotten (RTBF) in which cases we are only required to provide security protection for the given target token sequences in the training data. Thus, evaluation scenarios such as membership inference attack recall are not suitable, but instead, targeted extraction attacks that assume adversaries have some prior information (prefix) are more suitable.

This also ties in with why we do not consider other traditional DP algorithms (e.g. DP-SGD) based on gradient-descent methods as baseline approaches, since these approaches are concerned with providing a general privacy guarantee **during** initial training. However, we are concerned with when individuals practice their RTBF and explore **Post-Hoc** methods for a PLM that is already finished with the initial training and is vulnerable to extraction attacks. We have explicitly mentioned and clarified our main objective in the revised draft.

**Additional  Baselines**

To address reviewer concerns (mentioned by 3 out of 4 reviewers) about the lack of baselines, we have implemented another baseline approach that can be used post-hoc pretraining w/o full re-training: a Differential Privacy (DP) decoding strategy for LLMs [2]. The method proposes ‘softening’ the probability of generating each token from the vocabulary and randomly sampling using the logits as sampling weights, which theoretically guarantees Differential Privacy. Thus, we consider [2] as a baseline approach.

Below is the summary of baseline results. We also add 4 generation tasks (dialogue) for evaluation since the baseline approaches (DP decoding) are not suitable for the existing classification tasks with a verbalizer.

| Model     | Size |  EL  |  MA  |  WoW |  ED  |  BST |  WoI |  Dialogue AVG |
|-----------|------|:----:|:----:|:----:|:----:|:----:|:----:|:----:|
| NEO       | 125M | 30.9 | 77.4 |  8.4 |  8.4 |  9.6 | 11.2 |  **9.4** |
| NEO + DP+  | 125M |  0.0 | 27.3 |  7.3 |  6.9 |  6.8 |  8.0 |  *7.3* |
| NEO + UL+  | 125M |  1.0 | 27.4 |  2.5 |  2.9 |  2.1 |  2.8 |  2.6 |
|-|-|-|-|-|-|-|-|-|
| NEO       | 1.3B | 67.6 | 92.2 |  9.6 | 10.5 | 12.2 | 13.7 | **11.5** |
| NEO + DP+  | 1.3B |  0.0 | 21.4 |  6.9 |  6.6 |  6.7 |  8.0 |  7.1 |
| NEO + UL+  | 1.3B |  1.9 | 30.4 |  9.0 |  7.2 |  8.0 |  9.7 |  *8.5* |
|-|-|-|-|-|-|-|-|-|
| NEO       | 2.7B | 70.4 | 93.4 |  9.2 | 10.9 | 12.4 | 13.6 | **11.5** |
| NEO + DP+  | 2.7B |  0.0 | 24.2 |  6.9 |  6.4 |  6.6 |  7.6 |  6.9 |
| NEO + UL+  | 2.7B |  1.6 | 31.0 | 11.4 |  9.9 | 10.8 | 12.2 | *11.1* |

Results show that while DP Decoding enables effective protection against extraction attacks demonstrated via the low EL and MA score, it results in severe degradation of generation capabilities measured via the Average F1 score of the 4 dialogue generation tasks. Knowledge unlearning also results in severe degradation for the smaller LM sizes. However, for 2.7B LM, it results in very little performance degradation, bolstering our initial finding that large LMs are better forgetters. We have included the new baseline and results in the revised draft.

---

> ### Author Response · Authors · 2022-11-15
> **Continued General Response**
>
> **Effect of hyperparameter ‘n’ in proposed EL metric (Reviewers 4t9e, 9RUh)**
>
> The n can be a proxy for measuring when an extraction attack is successful. For example, a very conservative approach might regard an extraction attack to be successful if two consecutive token sequences of the target token sequence (length of 200) are extracted; then the n is set to 2. Likewise, we set the n value to 10 since we empirically consider an extraction to be successful when 10 consecutive token sequences are successfully generated by the LM (SSN, Phone Numbers, etc., are within the 10 token range). We show the result of varying n with values [5,10,20,40] for the GPT-Neo 1.3B LM when forgetting 32 samples (s=32) to the threshold value.
>
> First, we show the EL Forgetting Threshold values for n=[5,10,20,40] by measuring the value on the 10,000 validation instances unseen during training.
>
> | Model (Size)   | EL-5 | EL-10 | EL-20 | EL-40 |   MA  |
> |----------------|:----:|:-----:|:-----:|:-----:|:-----:|
> | GPT-Neo (1.3B) | 7.85 |  5.68 |  4.07 |  2.66 | 33.27 |
>
> Next, we show the average LM performance (on the 9 classification benchmarks) where we perform unlearning on the 1.3B LM on 32 samples until the target token sequences are forgotten (the EL & MA become lower than the threshold values). Performance shows the average of 5 random samplings.
>
> | EL-n  | AVG (Acc) |
> |-----------------|-----------|
> | EL n=5   | 49.93     |
> | EL n=10 | 49.93     |
> | EL n=20 | 49.85     |
> | EL n=40 | 49.88     |
>
> As shown by the results, the effect of varying the hyperparameter ‘n’ is minimal on the final LM performance.
>
> **Final Authors' Note**
>
> In our revised draft, we have provided an additional baseline based on a DP decoding strategy [2] that can also be applied w/o full re-training. We have also provided clarification for the evaluation setting, addressing two of the main common concerns raised by the reviewers.
>
> As the reviewers have also commented, we believe that our work is the first to benchmark initial results for exploring unlearning for large language models, which is timely considering the rapid and untested application of large language models in regard to individuals' privacy. We show that simply performing gradient ascent without any additional mechanism can be a strong baseline for providing protection against targeted extraction attacks, especially as LMs scale to larger sizes. We believe our work can be the starting point for future work exploring efficient privacy mechanisms for large language models, and we kindly ask the reviewers to reconsider their initial assessment.
>
> **References**
>
> [1] Kushal Tirumala, Aram H Markosyan, Luke Zettlemoyer, and Armen Aghajanyan. Memorization without overfitting: Analyzing the training dynamics of large language models. arXiv preprint arXiv:2205.10770, 2022
>
> [2] Majmudar, Jimit, Christophe Dupuy, Charith Peris, Sami Smaili, Rahul Gupta, and Richard Zemel. "Differentially Private Decoding in Large Language Models." arXiv preprint arXiv:2205.13621 (2022).
>
> [3] Li, Xuechen, Florian Tramer, Percy Liang, and Tatsunori Hashimoto. "Large language models can be strong differentially private learners." ICLR 2022.
>
> [4] Anil, Rohan, Badih Ghazi, Vineet Gupta, Ravi Kumar, and Pasin Manurangsi. "Large-scale differentially private bert." arXiv preprint arXiv:2108.01624 (2021).

---

### Decision · Program_Chairs · 2023-01-20

**Decision:**

Reject

**Justification For Why Not Higher Score:**

See the aforementioned main concerns (A) and (B).

**Justification For Why Not Lower Score:**

N/A.

**Metareview: Summary, Strengths And Weaknesses:**

The main contribution of this work lies in proposing the use of gradient ascent on equation 1 for unlearning to provide empirical privacy guarantees for large language models.

This paper has generated considerable response to the authors' rebuttal and an active discussion from the reviewers.

While we have found the findings interesting, two main concerns remain after reviewing the authors' responses:

(A) It is unclear if the authors' claim of "the simple approach of gradient ascent ... results in little to no degradation of general performance for LMs" would indeed hold when larger datasets are unlearned. In particular, when would be the "breaking point"? Reviewers 9RUh and 4t9e have both raised questions regarding this, but have not received a satisfactory response. Answering this question would allow us to better understand concretely the extent and specificity of this claim.

(B) Related to A, it is not clear why unlearning a larger batch of examples (128 vs 32) performs significantly worse than sequentially unlearning smaller batches of 32 examples at a time. The authors were not able to give a convincing explanation. Addressing this concern will require the authors to perform a careful detailed investigation.

The authors are strongly encouraged to revise their work based on the above and the reviewers' other feedback and suggestions.

**Summary Of Ac-Reviewer Meeting:**

As mentioned in the meta-review, the reviewers in attendance have raised and agreed on the above main concerns (A) and (B), which resulted in the final recommendation/decision.